# Spatial summation of pain is associated with pain expectations: Results from a home-based paradigm

Jakub Nastaj[1], Jacek Skalski[1], Aleksandra Budzisz[1], Tibor M. Szikszay[2], Sylwia Swoboda[1], Weronika Kowalska[1], Daria Nowak[1], Edyta Zbroja[1], Natalia Kruszyna[1], Marta Jakubińska[1], Dominika Grygny[1], Romuald Polczyk[3], Andrzej Małecki[1], Kerstin Luedtke[2], Wacław M. Adamczyk[1]*

**1** Laboratory of Pain Research, Institute of Physiotherapy and Health Science, The Jerzy Kukuczka Academy of Physical Education, Katowice, Poland, **2** Institute of Health Sciences, Department of Physiotherapy, Pain & Exercise Research Lübeck, Universität zu Lübeck, Lübeck, Germany, **3** Institute of Psychology, Jagiellonian University, Kraków, Poland

* w.adamczyk@awf.katowice.pl

**Data Availability Statement:** All relevant data are within the manuscript and its Supporting Information files.

## Abstract

The purpose of this study was to reproduce the previously observed spatial summation of pain effect (SSp) using non-laboratory procedures and commercial equipment. An additional aim was to explore the association between expectations and SSp. The Cold Pressor Task (CPT) was used to induce SSp. Healthy participants (N = 68) immersed their non-dominant hands (divided into 5 segments) into cold water (CPT). Two conditions were used 1) gradual hand immersion (ascending condition) and 2) gradual hand withdrawal (descending condition). Pain intensity was measured on a Visual Analogue Scale (VAS). Psychological factors, such as the participants' expectations of pain intensity were also measured on a VAS. Results showed significant SSp ($\chi^2_{(4)}$ = 116.90, $p$ < 0.001), reproduced with non-laboratory equipment in a home-based set-up. Furthermore, two novel findings were observed: i) there was a significant correlation between expectations and perceived pain, indicating a link between pain expectations and SSp, ii) spatial summation increased with the increase in duration exposure to the noxious stimulus (Wald $\chi^2_{(8)}$ = 80.80, $p$ < 0.001). This study suggests that SSp is associated with pain expectations and can be formed by a mixture of excitatory and inhibitory mechanisms potentially driven by temporal characteristics of neural excitation. Moreover, this study proposes a new feasible way to induce SSp using a home-based set-up.

## Introduction

Spatial summation of pain (SSp) is characterized by an increase of perceived pain intensity when the area of nociception is enlarged [1–3]. Furthermore, pain thresholds decrease as the area of noxious stimulation increases [4–6] which can also be seen as a spatial summation manifestation. There are also studies indicating that SSp can occur when the area of

**Funding:** WA is supported by the National Science Center within the grant no. 2020/37/B/HS6/04196. The funders had no role in study design, data collection and analysis, decision to publish, or preparation of the manuscript.

**Competing interests:** The authors have declared that no competing interests exist.

nociception is not contiguous [7, 8]. However, studies show that when the separation between stimulated areas exceeds 30 cm (in case of cold pain) [9] or 10 cm (in the case of heat pain) [10], SSp no longer occurs. Several possible mechanisms have been proposed to contribute to the SSp effect, e.g., local stimulus integration [11], neural recruitment [8, 11], lateral inhibition [12], or diffuse noxious inhibitory control [3]. An interesting observation is that there is no exponential increase in pain intensity during a linear increase in the "size" of the stimulated area [1], a typical stimulus-response pattern when the stimulus area (size) is constant but the noxious intensity increases linearly [13, 14].

The cold pressor task (CPT) is widely used in studies on nociception [2, 15–20] and SSp [2, 18–23]. In CPT, a body part is immersed into cold water serving as a noxious stimulus. The low temperature activates nociceptive fibers [20, 24, 25], probably through low temperature sensitive ion channels TRPM8 [26] and TRPA1 [27] leading to increasing pain of mild to moderate intensity [28]. Although standardization in this method is difficult due to e.g., anthropometric differences, it is easy and commonly used and has been successfully employed in pain experiments. A similar method using hot water instead of cold, was also used by Marchand & Arsenault [3] to study SSp. Their results showed that SSp was observed only in the condition using a gradual decrease of the stimulated area but did not occur in a progressively increased stimulated area. Moreover, the perceived pain was less intense during the decreasing compared to the increasing condition.

One of the factors that can be related to SSp are expectations [29, 30]. Studies on stimulus expectations in pain showed that even short-term predictive cues could have effects on pain perception [31, 32]. The role of the expected (predicted) value on pain perception can be explained using the predictive coding framework [33, 34], wherein the prior distribution of predicted values leads to a shift of value distribution after the integration of the collected data, in this case nociceptive information. Interestingly, despite the widely documented effect of expectation on pain perception, the relationship between expectation and SSp has not yet been studied.

Publication bias, based on the phenomenon of a higher probability of publishing statistically significant findings than nonsignificant findings [35], and the lack of replication studies, are affecting the validity of scientific research [36]. To address this problem in the context of SSp research, one of the aims for the current study was to replicate the SSp effect, adopting the methodology from the study published by Marchand & Arsenault [3]. Based on previous studies, applying the CPT paradigm using non-laboratory equipment without water circulation [37, 38], the authors decided to additionally test if it is possible to conduct and replicate, SSp outside the laboratory setting [23]. This novel, yet adopted methodology could enable the conduction of similar replications and preliminary studies for researchers who do not have dedicated equipment for CPT in their laboratories, or it could be used in exceptional situations when conducting research in laboratories is hampered [39, 40]. Moreover, the proposed method could be very useful for clinical purposes if there is lack of dedicated CPT equipment. Furthermore, the current experiment aimed to investigate the potential association between pain-related expectations and SSp by examining participants' predictions regarding pain using a SSp paradigm. It was hypothesized that participants can predict pain produced by noxious stimulation of different sizes.

## Materials and methods

### General information

The study design was based on the experiment by Marchand & Arsenault [3]. The study was designed as a within-subject experiment. Each participant took part in two consecutive

experimental conditions during which they immersed their hand into a cold-water tank. The study was conducted in a home environment, i.e., each examiner performed an experiment within their households (see details below). The home setting differed between examiners. Mainly living rooms, bedrooms and kitchens were chosen for the setup. Settings were chosen to reduce distractions such as music or the sounds of other household members. Examiners followed an intensive training for the precise data collection and screening procedures. Each examiner was involved in all phases of study preparation. Firstly, they took part in a series of online meetings during which they were introduced to the design of the experiment. Secondly, they received training on the preparation of the CPT (see below). During training, water temperatures achieved in their home environment were measured (water temperature in CPT before adding 6 foil ice-cube packs and after adding ice cubes every 10 minutes for 2h). Thirdly, examiners conducted a pilot study, which was preceded by training in the laboratory. Each of the examiners was individually trained how to follow each step of the study SOPs (standard operating procedures) and was supervised by JN. Each examiner conducted a pilot assessment of one single individual in their home environment. To investigate if the CPT method was consistent among examiners, measures of temperatures were compared (see S1 Table). Finally, before conducting the main experiment, examiners demonstrated all procedures to JN, who assessed them in terms of quality and ensured feedback if any violations from SOPs were detected. The study was approved by the Bioethics Committee of the Academy of Physical Education in Katowice (1-2021/02-25-21) and was conducted in accordance with the Declaration of Helsinki [41]: Every participant provided written informed consent. The study was pre-registered at the OSF platform (*https://osf.io/kjbdz*). Protocol deviations are described in the S1 Text.

## Study population

Examiners (n = 13) consisted of members of the Laboratory of Pain Research at the Academy of Physical Education in Katowice or laboratory collaborators. Only healthy participants aged 19–65 years could take part in the study. A thorough interview was conducted using a screening questionnaire which allowed to enroll healthy participants [42]. Participants were excluded if they had a current trauma or wounds in the non-dominant hand, had COVID-19 disease (at the study date or in the past), suffered from acute pain on the study date or within 24 hours preceding the study, took psychoactive substances or medications on the study date, were diagnosed with a disease related to cold temperature intolerance (e.g., Raynaud syndrome, cryoglobulinemia, cold urticaria, etc.), had experienced in the past a pathological reaction to cold temperature (e.g., excessive edema or redness, blisters, etc.). Additionally, because the experiment was conducted outside the laboratory and to avoid any adverse events, rigorous exclusion criteria were applied (see S2 Text). Apart from those derived from previous experiments on cold pressor task (see [43] for example), additional exclusion criteria were obtained from the literature regarding cryotherapy [44, 45]. Lastly, if there was any concern about the health condition, the decision to participate was made by a medical doctor who was a member of the research team (AM).

## Equipment for the cold pressor task

Because the experiment was designed to be performed in a non-laboratory setting, commercially available equipment and tools were used. Transparent plastic rectangular containers (36.5 × 27.5 × 22cm) filled with cold tap water (15cm height of the container) were used for the Cold Pressor Task (CPT). To obtain the desired water temperature of approximately 5˚C, 6 foil ice-cube packs were used (a total of 144 ice cubes) for each experimental condition. An

electronic (±0.1˚C) thermometer was attached to the plastic box [23] to monitor the temperature. This non-laboratory version of the CPT was previously validated and led to comparable SSp induced via laboratory-based cold pressor with water circulation [23].

## Experimental design

Before the first immersion of the hand, participants were instructed about the test procedure and prepared for the measurements. Subsequently, the participants were asked about their general fear of pain and fear of pain caused by the cold temperature by using a 0–10 Verbal Rating Scale, where 0 was defined as "no fear of pain" and 10 as "the greatest fear of pain you can imagine". The non-dominant hand was then divided into 5 segments using the anatomical points of the hand (Fig 1): (1) first segment - from the fingertips to the distal interphalangeal joint of the third finger; (2) second segment - from the distal interphalangeal joint to the proximal interphalangeal joint of the third finger; (3) third segment - from the distal interphalangeal joint of the third finger to the metacarpophalangeal joint; (4) fourth segment and (5) fifth segment - the distance between the flexion line of the metacarpophalangeal joint of the third finger and the beginning of the flexion line of the metacarpophalangeal joint of the thumb divided into two parts. Segment lines were marked on the skin with a pen. Length (a) and width (b) of each segment was measured using flexible measuring tape before the main phase of experiment. To calculate the "area" of immersed segments, surface areas ($a_x \times b_x$) of each immersed segment were summed. For example: the surface area of 5/5 segments was calculated as: first segment ($a_1 \times b_1$) + second segment ($a_2 \times b_2$) + third segment ($a_3 \times b_3$) + fourth ($a_4 \times b_4$) fifth segment ($a_5 \times b_5$). The size of areas exposed to cold stimulation is presented in Fig 2. Before and during the main phase of the experiment, examiners recorded the water temperature at multiple time points as well as the room temperature before each experimental condition. The amount of ice added during the experiment was also recorded. Experiments began when the water temperature had decreased to 4.5–5.5˚C. Two experimental conditions, ascending and descending, were used for all participants. In the ascending condition (see Fig 1), participants started from immersion of 1 segment and sequentially immersed a greater number of segments, finishing with all 5 segments immersed. In the descending condition the order was reversed, i.e., they first immersed all 5 segments and finished with immersion of just 1 segment. In both conditions there was a 5 minutes break between immersion of the next segment to ensure that the skin temperature returned to baseline [21]. The order of testing (ascending vs. descending condition) was random. The interval between each condition was one hour. During that break, participants were asked to complete the SEWL (subjective experience of workload) questionnaire which measures physical activity level [46, 47].

Before and after each experimental condition (ascending, descending), cold pain thresholds ($PT_{COLD}$) were tested on the examined limb: an ice cube was placed on the palmar surface of the participants' hand and the time (in seconds) until the first pain sensation occurred, was recorded [48]. This procedure was used to control for cold sensitivity and was an integral part of the screening procedure. At the end, subjects were asked to provide demographic information and to guess the purpose of the study. None of participants knew the correct purpose of the experiment. A familiarization phase was not embedded within the study design to not bias expectations regarding pain from cold water immersion. However, measuring $PT_{COLD}$ prior to the main data collection, allowed participants to familiarize with the cold sensation without inducing expectations. A detailed presentation of the study flow is shown in S1 Fig.

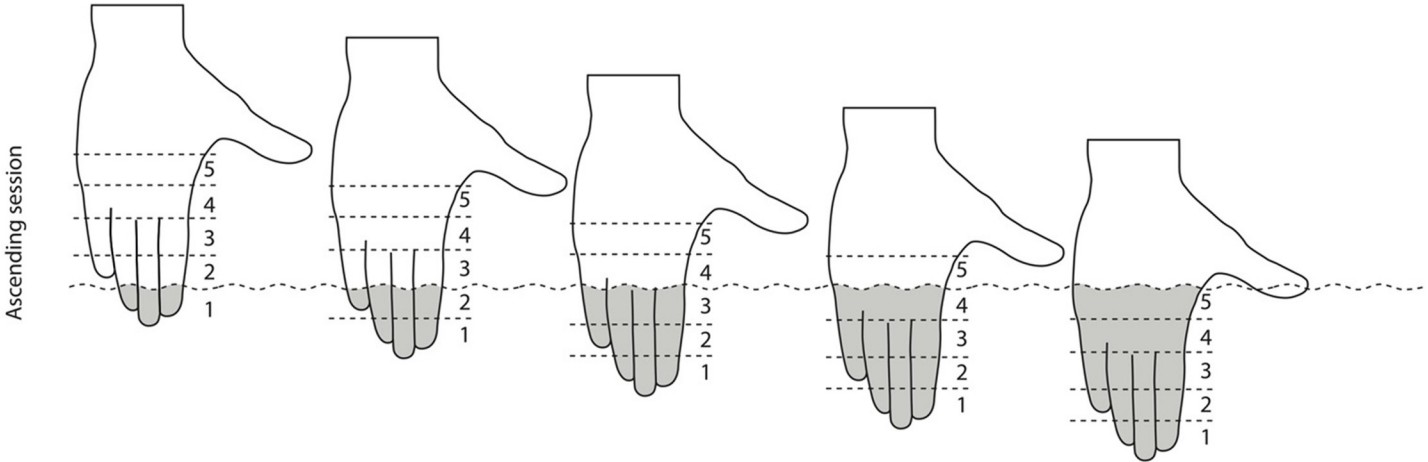

**Fig 1. Experimental procedure.** Demonstration of the "ascending" condition with the hand divided into 5 segments: In the ascending condition, nociceptive stimulation started from a small area of the hand (fingertips) and increased in subsequent immersions. In the descending condition, the order was reversed: nociceptive stimulation started with the whole hand immersed (segments 1 to 5) and decreased number of segments in subsequent immersions.

### Trial design

Each immersion/trial lasted 60 seconds (regardless of the number of segments involved) and inter-trial intervals were set at 5 minutes [3, 22]. Each single trial started with a question about expected pain intensity for this trial. Participants were told and shown a figure demonstrating the area of the hand which will be exposed to water immersion. Participants were asked to mark a point on the Visual Analogue Scale (VAS) which best reflected their expected pain intensity. The scale for expectations had anchors of 0 (no pain) to 100 (worst pain imaginable). The first measurement of expectation (prior to any immersion) started with the descending condition of segments 5/5 (segments 1–5). Next, participants were instructed to immerse their hand up to the line which separated a given number of segments (Fig 1). They were explained and shown to stabilize their thumb to avoid its accidental stimulation. Participants were prompted to rate their pain intensity on the VAS scale (same as for expectations) at the following time points: after 10, 30 and 50s. Participants were blinded towards their previous ratings as these were covered and remained inaccessible during the study. While participants provided ratings, examiners recorded the temperature of the water in a room where assessments were run. After removing their hand, participants were instructed to rest their hand in their axilla for approximately 4 minutes and 30 seconds. This allowed the skin temperature of the hand to return to the baseline level before the next immersion.

### Data extraction and statistical analysis

The main analyses were conducted in the following stages: In the first stage, the effect of stimulation area on pain intensity was investigated using a General Estimated Equations (GEE) model with three within-subject conditions treated as "factors": "condition" (ascending, descending), "segment" (1, 2, 3, 4, 5 segments) "time" (10s, 30s, 50s). The same analysis was repeated with pooled temperature set as a covariate. Polynomial contrast analysis was performed to check the pattern (nonlinear, linear) of pain increase between immersions. Furthermore, in case of significant main and/or interaction effects, *t*-tests contrasts were performed to describe reported effects.

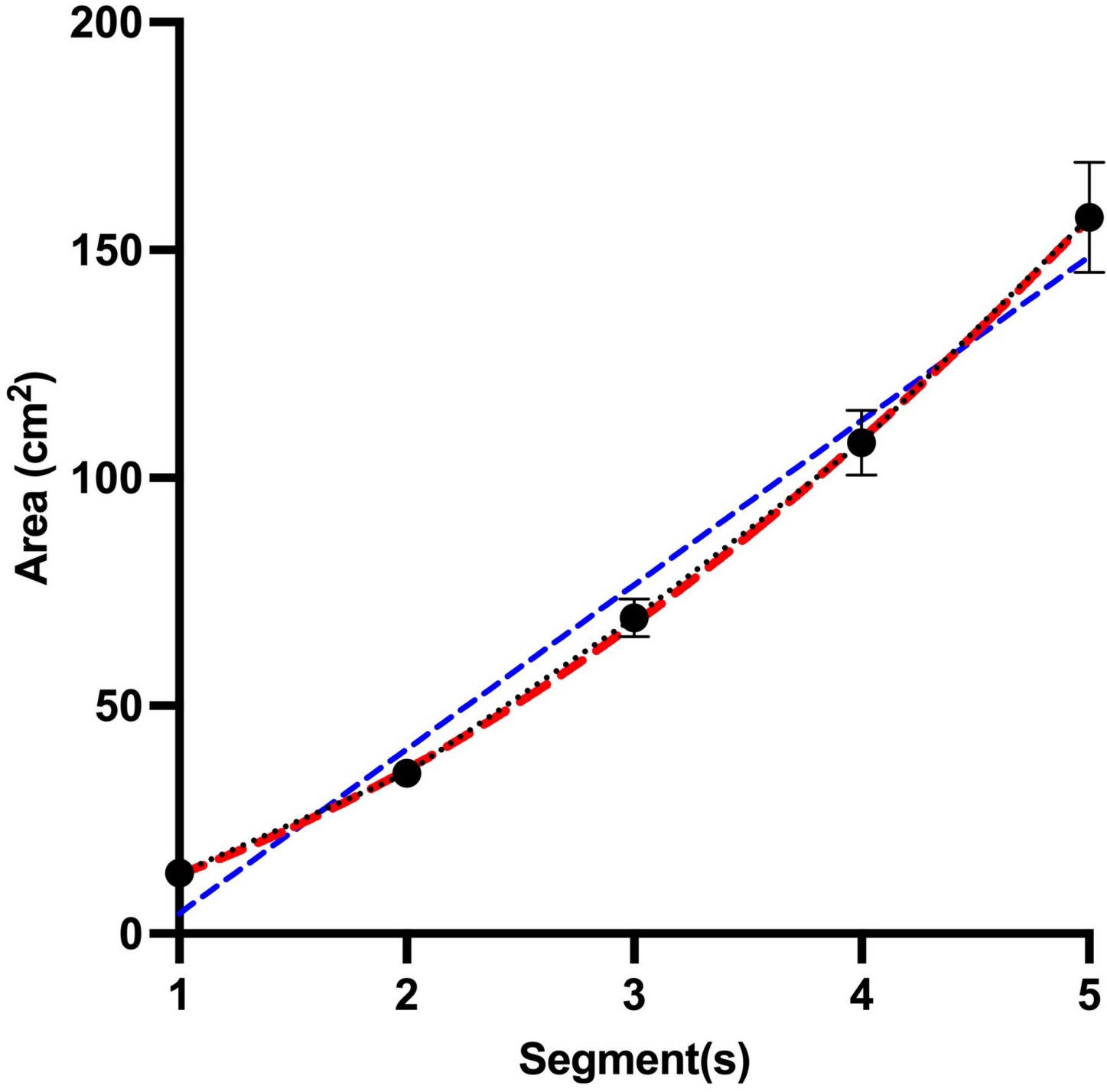

**Fig 2. Hand areas exposed to noxious cold stimulation.** Exponential (red) growth negligibly better ($R^2$ = 0.48 vs. 0.47) than linear (blue) fit the increase of the area of stimulation ($p < 0.001$).

Next, two sets of expectation ratings (measured before vs. after) were compared within the same statistical model to explore potential differences. The effect of stimulation area on pain expectations was tested using (GEE) with two within-subject factors: "condition" (ascending, descending) and "segment" (1, 2, 3, 4, 5 segments). Next, the differences in the actual sizes of

areas of stimulation (approximated via anthropometric measurements) were analyzed by using repeated measures ANOVA and size of immersed segments (1,1+2, 1+2+3, 1+2+3+4, 1 +2+3+4+5) as dependent variables. Greenhouse-Geisser correction was applied if sphericity assumption was violated. Linear regression was calculated to describe the relationship between pain and number of stimulated segments–similarly to the described procedures in the Marchand & Arsenault study [3].

Lastly, the correlation between pain intensity and pain expectation was conducted using Pearson product-moment coefficients, scatter plots and regression lines fitted to the data. Correlation between SSp (difference between pain in immersion of 5 segments versus 1 segment) and physical activity (subjective experience of workload [SEWL] score), age, Δarea (difference in size between all 5 segments and 1 segment), and temperature of the water, were also tested. The comparison between water temperature and $PT_{COLD}$ before and after the experiment was conducted using paired Student $t$ tests. The α level was set as 0.05 and Sidak correction was employed to control type-I error rate. Unless otherwise indicated, figures and tables include descriptive statistics based on raw data. All statistical analyses were performed using the IBM Statistical Package for Social Science (SPSS version 25, Armonk, NY, USA).

## Results

A total number of 68 participants (29 males; 39 females; age: 18–57, mean: 31.08) completed the experiment. Further descriptive characteristics are shown in Table 1. The mean temperature of the cold water was 5.13˚C during the ascending condition and 5.12˚C during the descending condition. In the ascending condition ($t_{[66]}$ = -0.5, $p$ = 0.63) as well as during the descending condition ($t_{[66]}$ = 1.2, $p$ = 0.22) no significant differences (before condition vs. after condition) were shown for the pain thresholds, indicating that pain sensitivity was stable over the course of experiments (S2 Table).

### Primary analyses

Means and standard deviations for pain are presented in S3 Table. The GEE showed a significant main effect for the factor "segment" (Wald $\chi^2$(4) = 116.90, $p$ < 0.001), indicating a significant SSp effect. Pairwise post-hoc comparisons showed significant differences in pain between the immersions of different numbers of hand segments. SSp occurred between segment 1 and segments 1+2 (Mean difference (MD): -4.1; 95% CI -6.84, -1.36), segment 1 and segments 1+2 +3 (MD: -9.88; 95% CI -13.66, -6.09), segment 1 and segments 1+2+3+4 (MD: -17.17, 95% CI

**Table 1. Descriptive statistics.**

| Variable | Mean (SD) |
|---|---|
| Age (years) | 31.08 (12.17) |
| Height (cm) | 173.83 (9.50) |
| Body mass (kg) | 68.23 (14.75) |
| Fear of Pain General (NRS 0–10) | 4.38 (2.15) |
| Fear of Pain Cold (NRS 0–10) | 3.36 (2.19) |
| SEWL | 10.31 (1.27) |
| **Variable** | **N** |
| Sex | F = 39 (57%); M = 29 (43%) |
| Handedness | R = 67 (98.5%); L = 1 (1.5%) |

F- Female, M -Male, R - right, L - left. NRS - numeric rating scale, SEWL - subjective experience of workload questionnaire, SD–standard deviation

-22.01, -12.33) as well as between segment 1 and segments 1+2+3+4+5 (MD: -23.72, 95% CI -29.95, -17.49). Furthermore, a linear relationship of these areas was shown as indicated by the polynomial contrast ($p < 0.001$) and regressions coefficients: Slope of pain increase was steeper in the descending compared to the ascending condition, however, the most noticeable difference in steepness was found for pain measured at 50s (B = 9.45 vs. 6.20, see S4 Table)

Furthermore, a significant main effect was found for the factor "time" (Wald $\chi^2(2)$ = 157.45, $p < 0.001$), indicating that (likely) temporal summation occurred during immersion. Pairwise comparisons showed a significant difference between the first (10s) and second (30s) as well as between the first and third (50s) (last) pain measurement ($p < 0.001$). No significant effect of "condition" (Wald $\chi^2(1)$ = 1.07, $p = 0.30$) was found.

Significant two-way interactions were found between the factors "condition" × "segment" (Wald $\chi^2(4)$ = 18.57, $p = 0.001$), "condition" × "time" (Wald $\chi^2(2)$ = 8.31, $p = 0.02$) and "segment" × "time" (Wald $\chi^2(8)$ = 80.80, $p < 0.001$). Pairwise comparisons following these effects revealed that there was a significant difference in pain between the ascending and descending condition but only for the immersion of segment 1 (Fig 3) ($p = 0.016$). Likewise, for the two-way "segment" × "time" interaction, pairwise comparisons showed that spatial summation was significant regardless of the timepoint of measurement, however, largest effects were observed for the last pain measurement (50s) ($p < 0.001$).

A significant three-way, i.e., "condition" × "area" × "time" interaction was shown (Wald $\chi^2(8)$ = 17.49, $p = 0.03$), indicating that different spatial summation trajectories were observed across the two conditions (ascending vs. descending) in respect to timepoint of measurement (10, 30, 50s). Exploration of this interaction with pairwise comparisons showed that significant differences between the ascending and descending conditions occurred for the immersion of segment 1, only during the third (50s) timepoint of measurement ($p = 0.017$) (Fig 4), but not for the second (30s) ($p = 1.00$) and first (10s) ($p = 0.23$). Adding "temperature" as a covariate had only a marginal effect on the three-way interaction (Wald $\chi^2(8)$ = 15.3, $p = 0.054$) and had no influence on other statistical results.

Descriptive statistics for expectation are depicted in Fig 3 and Table 2. No significant difference was observed between expectation measured before the first immersion (unbiased) and expectation measured during the assessment (Wald $\chi^2(1)$ = 1.21, $p = 0.27$). Thus, for clarity reasons, expectation measured during the assessment is presented below, whereas measurements taken prior to the assessment are detailed in S5 Table. GEE on expectations data did not show a significant main effect for the factor "condition" (Wald $\chi^2(1)$ = 3.04, $p = 0.08$), but for the factor "segment" (Wald $\chi^2(4)$ = 111.89, $p < 0.001$). Pairwise comparisons showed significant differences in expected pain between immersions of different numbers of hand segments (all $p$ values $< 0.001$). Namely, the expected pain level was lower for a single segment compared to two (MD): -4.95; 95% CI -7.58, -2.32), three (MD: -13.00; 95% CI -17.60, -8.40), four (MD: -19.38, 95% CI -24.92, -13.84) as well as five segments (MD: -23.38, 95% CI -29.82, -16.95). The "segment" × "condition" interaction was also significant (Wald $\chi^2(4)$ = 27.30, $p < 0.001$), indicating that the pattern of increase in expected pain level was different across the two conditions (Fig 3). Pearson coefficients ($r = 0.53–0.81$) were statistically significant for all correlations between expected pain and actual pain (Table 3 and Fig 5).

## Results of exploratory analyses

Exploratory correlations revealed no significant relationship between the magnitude of SSp and physical activity ($r = -0.04$, $p = 0.72$), maximal increment in the stimulated area (total stimulation area for 5 segments minus only 1 segment, $r = -0.14$, $p = 0.26$), or mean water temperature ($r = -0.11$, $p = 0.36$), as well as age ($r = -0.03$, $p = 0.82$). Further, repeated measures

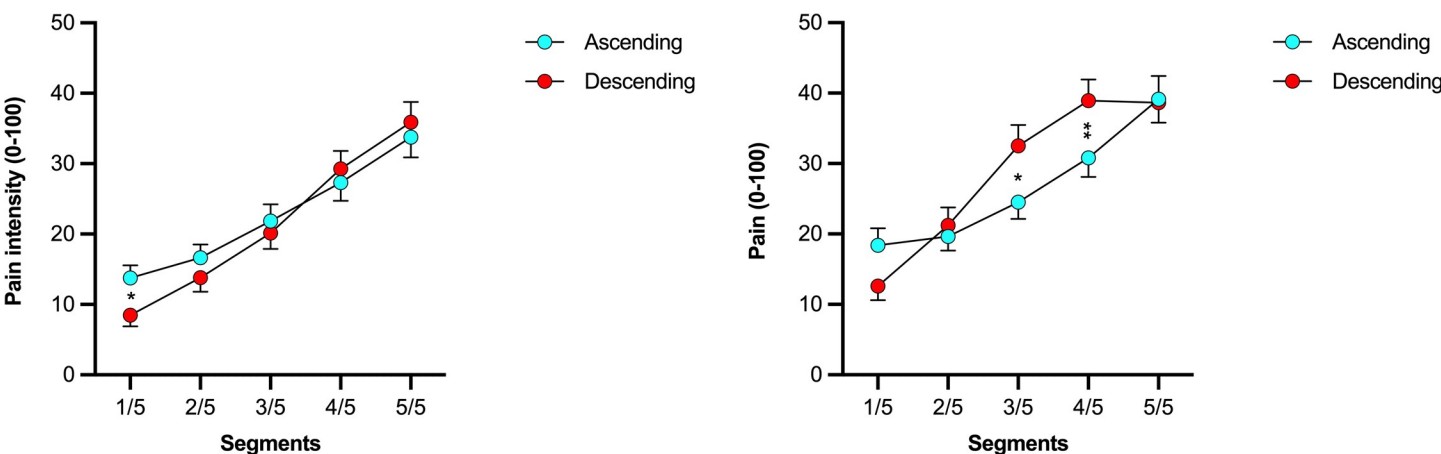

**Fig 3. Significant interactions between segment immersions and condition.** Left: Estimated marginal mean pain ratings during immersion of each number of segments for the two conditions (ascending, descending). Right: Mean expectations ratings prior to the first immersion of each number of segments for the two conditions. *Significant difference between the ascending and descending conditions for segment 1 ($p = 0.015$). **Significant difference in expected pain level ($p < 0.01$) between the ascending and descending conditions.

ANOVA of the size of immersed segments showed a significant effect for "size", indicating that the area of stimulation increased from trial to trial ($F_{(1.08, 72.17)} = 135.17$, $p < 0.001$, $\eta^2_p = 0.70$). Polynomial contrasts showed that the increase in size of the stimulated area can be explained by a linear and an exponential function ($F_{(1,67)} = 35.47$, $p < 0.001$, $\eta^2_p = 0.35$).

## Discussion

The main aim of this study was to reproduce the SSp effect, reported previously using an ascending/descending paradigm [2, 3, 22] and noxious cold stimulation [2, 22]. The second aim was to introduce a novel methodology to study SSp in a non-laboratory setting. The final aim was to use this novel methodology to investigate the associations between expectations and SSp.

Current results confirmed that SSp, as previously shown with noxious cold [22], and heat [3] stimulation of the upper extremity, can be reproduced in a home-based setting. The current findings imply that the proposed "adapted" paradigm is feasible for conducting bedside testing in clinical and non-laboratory environments. Furthermore, these current results contribute to the mechanistic understanding of SSp by showing that spatial summation might be predicted by subjects' pain expectations. Moreover, enhanced SSp over time may suggest that this effect is potentially driven by temporal characteristics of neural excitation. Lastly, differential effects of the order of immersions indicate that -to some degree- SSp is shaped by descending pain inhibition.

### Reproducibility of spatial-related effects

The current study aimed to reproduce the effect previously shown by Marchand & Arsenault [3], yet using a modified methodology. In the mentioned experiment, participants' upper extremities were divided into 8 segments, such that the first segment included only the fingertips and the last segment included the entire arm (from fingertips to axilla). They used three experimental sessions: increasing session (immersion from fingertip to the axilla), decreasing session (immersion from the axilla to fingertip) and whole-arm-first + increasing session (first immersion to the axilla and next from fingertip to the axilla). Authors not only observed a significant SSp effect, as pain was on average higher when a larger area was stimulated, but also

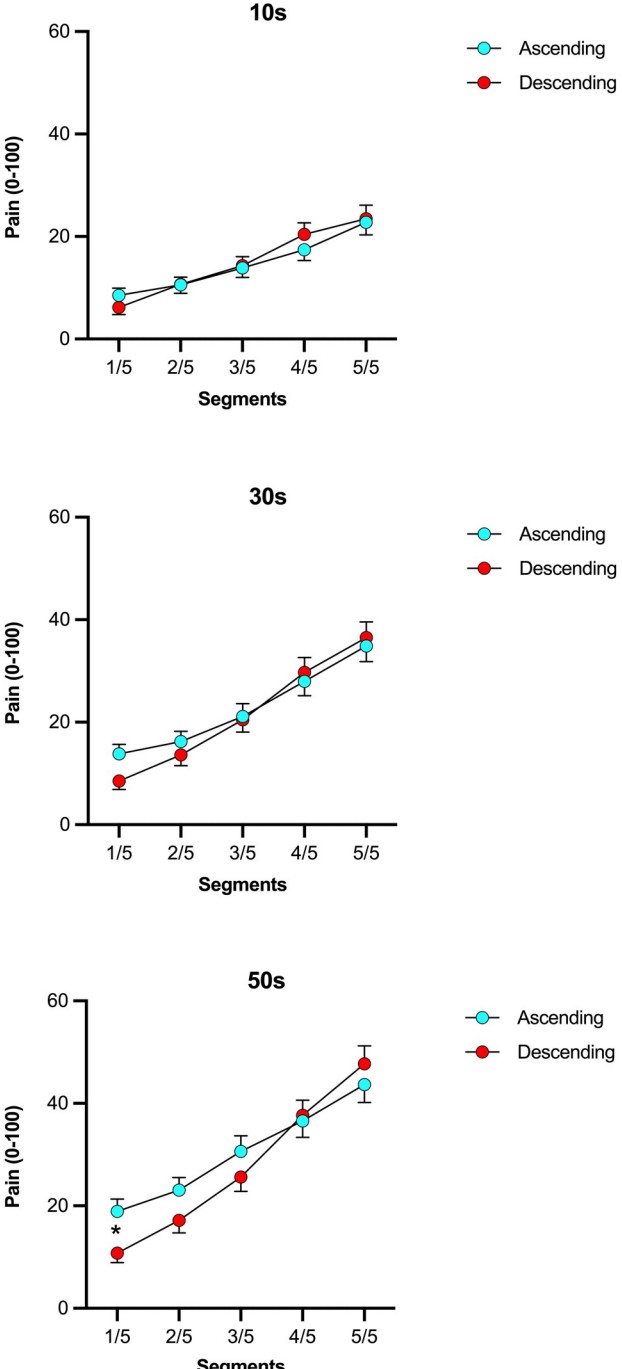

**Fig 4. Temporal summation drives the interaction between conditions (ascending, descending) and spatial summation of pain.** The figure shows mean pain ratings collected at 10s (3a), 30s (3b) and 50s (3c). Note that the difference in pain between ascending and descending conditions was significant ($p$ = 0.017) after 50s of immersion.

an interaction between the sequence of immersions and the size of the stimulated area. This interaction indicated that the same size of stimulated body area (segments) was perceived differently in the ascending (immersion from fingertip to the axilla) compared to the descending (from axilla to fingertips) condition. Their findings demonstrated that SSp was only detected

**Table 2. Pain-related expectations during the spatial summation paradigm.**

| Segment | Ascending | | | | Descending | | | |
|---------|-----------|------|------|-----|------------|------|------|-----|
| | Mean | SD | M | IQR | Mean | SD | M | IQR |
| 1/5 | 18.41 | 19.93 | 9.00 | 23 | 12.59 | 16.36 | 7.00 | 15 |
| 2/5 | 19.66 | 16.67 | 16.00 | 17 | 21.24 | 20.85 | 14.00 | 26 |
| 3/5 | 24.50 | 19.31 | 21.00 | 27 | 32.50 | 24.70 | 30.00 | 35 |
| 4/5 | 30.82 | 22.33 | 26.50 | 33 | 38.94 | 24.71 | 38.00 | 36 |
| 5/5 | 39.13 | 27.15 | 32.50 | 48 | 38.63 | 23.37 | 37.00 | 40 |

1/5 –Segment 1, 2/5 –Segments 1 to 2, 3/5- Segments 1 to 3, 4/5 –Segments 1 to 4, 5/5- Segments 1 to 5, SD, standard deviation, M, median. IQR, interquartile range.

in the decreasing session and whole-arm-first + increasing session and did not take place in the increasing session condition. The authors proposed an explanation that in the increasing condition, both facilitatory and inhibitory mechanisms were gradually being recruited. The results from the third condition (whole-arm-first + increasing session) appeared to support their suggestion. If the gradual recruitment of inhibitory mechanisms is responsible for the lack of a correlation between the stimulated area and reported pain during the increasing session, a positive relationship between the stimulated area and pain should be reestablished in the whole-arm-first + increasing session, whereas starting with the whole arm immersion would stimulate the activation of inhibitory mechanisms at the beginning of the session. In this current study, this interaction was replicated, although it was more prominent when only the immersion of the first segment (fingertips) was compared. In line with the previous observation, this interaction can be a manifestation of a robust activation of the descending pain inhibitory system during the descending condition.

Interestingly, SSp was reproduced in this current experiment in a cohort of 68 healthy individuals, despite introduced changes to the original methodology [3, 22, 49]. These differences include the localization of noxious stimulation as well as the overall size of the stimulated surface area. Only the hand was used for the current study and was divided into 5 segments, while Marchand & Arsenault [3] used the entire arm and divided it into 8 segments. Secondly, the type of noxious stimulation was different. Here, noxious cold stimulation was used, compared to noxious heat stimulation in the previous report. The type of noxious stimuli could explain the results as the most noticeable difference of pain increase between the two conditions was observed after 50s of immersion. This could be a result of temporal summation of noxious cold pain, which arises slower compared to noxious heat pain [50]. Thirdly, the study was conducted with home-based equipment outside of the laboratory. The latter aspect supports the robustness of observed findings. Despite larger variability and random noise cause by different CPT temperatures within the individual households, SSp was stable. Results were replicated even when controlling for water temperature.

**Table 3. Correlations between expectation and perceived pain.**

| Condition | Immersed segments | | | | |
|-----------|-------------------|-----|-----|-----|-----|
| | 1/5 | 2/5 | 3/5 | 4/5 | 5/5 |
| **Descending** | 0.85[***] | 0.76[***] | 0.79[***] | 0.82[***] | 0.65[*] |
| **Ascending** | 0.73[***] | 0.71[***] | 0.82[***] | 0.74[***] | 0.85[***] |

Correlation coefficients:

[***] $p < 0.001$

[*] $p < 0.05$

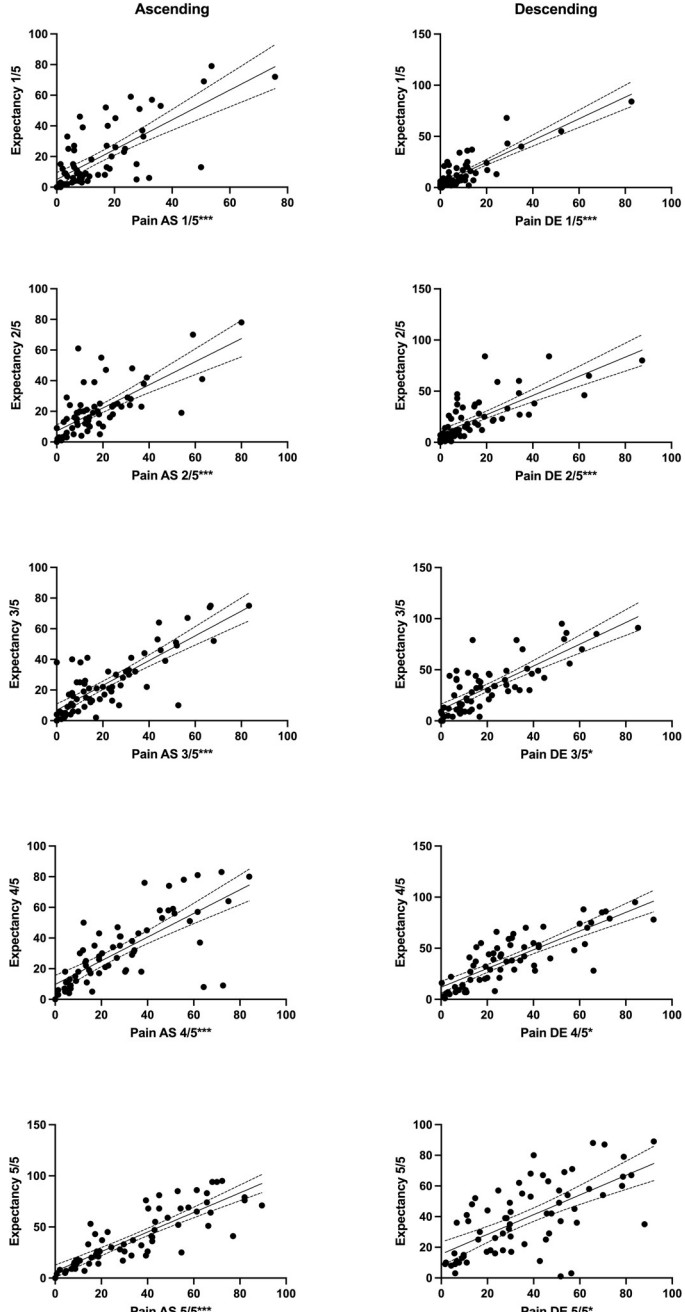

**Fig 5. Associations between predicted pain and actual pain of participants.** Associations between mean pain ratings during immersion for each number of segments in the two conditions and expectations measured during immersions into cold water. Left: ascending condition, right: descending condition. Significant correlation coefficients: $^*p < 0.05$, $^{***}p < 0.001$.

## Physiological mechanisms of SSp

The mechanisms of SSp are not fully understood, however, it seems that both facilitatory and inhibitory processes interact during summation. Inhibitory processes could be inferred from an interaction between the sequence of immersion and the stimulated body area (segments). It was observed that when the stimulation started from the largest area, pain provoked by

immersion of segment 1 was slightly lower than the analogue stimulation in the ascending sequence. That discrepancy can be explained by the fact that in the descending sequence, inhibitory mechanisms are activated to their maximum and persist during subsequent immersions, thereby resulting in lower pain during the immersion of segment 1, which is in line with a previous study [3]. Moreover, one study conducted on rats showed that an increase in stimulation area from 1.9 to 18cm$^2$, gradually decreased the frequency of convergent neuron discharges in intact, yet not spinally transected animals [51]. Furthermore, the increase in pain in either sequence was disproportional. A 5 times larger area does not multiply the reported pain intensity by the same value, which is in line with previous SSp studies [1, 3, 22].

As for facilitatory processes, it appears that they could be modulated by other factors. Temporal summation has not yet been considered in SSp induced via CPT. Our results suggest that the duration of the stimulus strongly affects the summation pattern. The curve representing the summation trajectory was steeper when the last measurements were considered (50s). During the immersion in cold water, the hand temperature had to change with time, hence, observed SSp could be driven by the gradual activation of deeply located primary afferents. This may indicate that SSp can be partially mediated by temporal and/or spatial summation of nociceptors innervating deeper layers of soft tissue [52] and possibly deep veins [53].

Another feature of SSp is the pattern of pain increase which was explained by a linear and a non-linear equation to the same extent. In a recent experiment with electrical stimuli, the pain increase followed a logarithmic curve when the size of the stimulated area increased in a linear fashion [1]. It can be hypothesized that in the current study, the increase in the stimulus area was rather exponential (Fig 4), which could result in a less efficient inhibition and thus a more linear increase in pain. However, this requires further research as a linear fit only negligibly less fit to these current data.

## Spatial summation using cold pain

Additionally to the topical administration of cold stimuli using a thermode [9, 54], SSp was previously reproduced by CPT in 6 experiments [18]; five of these showed significantly higher pain during the immersion of a larger area of the body [18, 19, 22, 23]. A first attempt by Wolf & Hardy [20] was not successful. The authors compared pain provoked by the immersion of one finger to pain provoked by whole hand immersion and did not observe SSp. However, the study sample was small (n = 2), and results could be explained by individual differences in SSp which are known to be large: lateral inhibition [12] may paradoxically lead to lower pain in a larger stimulated area [12, 21]. In a study by Westcott et al. [19], SSp was confirmed in 40 individuals by demonstrating more intense pain during full-hand immersion compared to the immersion of one finger into water of 0˚C. In one study, SSp was provoked with a temperature comparable to that used in the current experiment (4.7˚C) [18]. Participants withdrew their hand faster during stimulation of the whole hand compared to partial immersion. Julien et al. [22] divided the upper extremity into 8 segments and performed ascending and descending immersions with a water temperature of 12˚C, which was necessary to allow patients with chronic pain to tolerate the stimuli. They induced robust SSp and its disruption in the group of patients suffering from chronic pain. Two recent studies from our group confirmed the existence of SSp during hand immersion by dividing the hand into 5 segments [23], or 2 segments (ulnar and radial side) [21].

## Spatial summation in a non-laboratory setting

Assessment of spatial summation outside the laboratory has never been investigated apart from our pilot study [23]. The proposed methodology was inspired by difficulties with the data

collection during the COVID-19 pandemic. However, proposed methodology exceeds beyond pandemic use. For instance, it could be used to conduct replication and preliminary studies by researchers who do not have specialized equipment for CPT in their laboratories. In fact, experimental pain research often requires expensive equipment allowing for e.g. a control of the temperature of a stimulus during the experiment [28]. This methodology can also be helpful for assessing participants with CPT who cannot come to the laboratory e.g., due to their health condition. Furthermore, it can be discussed that the current study is a step forward by moving the laboratory-based pain research into field studies. This approach could be very useful in a clinical context when there is a lack of specialized equipment for CPT or there are difficulties to transport CPT equipment to patients who have musculoskeletal dysfunctions.

The methodology employed here was recently proposed in a preliminary validation study, conducted on 9 volunteers. This previous publication validated home-based CPT against laboratory-based equipment with a constant temperature and circulation of the water [23]. The pilot study showed that the SSp trajectory was comparable in the laboratory vs. the non-laboratory paradigm, although the average temperature of the water might have been higher and more variable in the home set-up. In another study by McIntyre et al. [55] healthy participants were trained (online) to self-administer CPT and showed that 97% of participants did not report issues with the test procedure of CPT. These findings together with the current results suggest, that the complex assessment of pain modulation can be used safely in the home environment.

### Expectations and spatial summation

Expectations have not yet been considered in SSp. Correlations reported in the current study were positive and significant in all cases, indicating that participants expected more pain before immersion of the larger area of their body. Participants correctly adjusted their expectations after each immersion and correctly predicted that a larger area of immersion caused more intense pain in the ascending condition. The same results were observed for the descending condition as participants correctly predicted a decrease in pain during subsequent immersion of a smaller area. This results could be explained by predictive coding [33, 56] as an increase or decrease in pain intensity during consecutive immersions was in line with participants' expectation.

In contrast to SSp, expectations -so far- were considered in conditioned pain modulation (CPM), a pain-inhibiting-pain effect. In a typical CPM experiment, participants are exposed to a test stimulus after the pre-exposure to a conditioning stimulus (CS). In one study [57], a high level of expected pain hampered the magnitude of the CPM effect. However, in a study by France et al. [58], expectations were matching the level of pain after application of the CS. Namely, those participants who expected lower reductions in CPM, experienced a more robust inhibitory effect. In another study, expectations were manipulated, showing that expectation -if influenced by verbal suggestion- lead to a reduced CPM effect - but only in females [59]. As both CPM and SSp are pain modulation paradigms testing the spatial aspects of pain, it is reasonable to assume that expectations also shape SSp.

### Limitations and future directions

There are two main limitations to this study. First is the lack of standardization of the environmental factors that result from conducting this study outside the laboratory settings. Examiners conducted the study in their households, resulting in significant differences in the characteristics of the selected rooms and environmental stimuli. The second limitation is that the study was conducted by multiple examiners, differing in factors such as age and gender

that can affect subject ratings. Another limitation is that the examiners varied significantly in the number of subjects assessed which comes as the result of restrictions during the COVID-19 pandemic. Unfortunately, a direct test of the influence of the examiners on the result was not possible, because of the large number of examiners. Each examiner tested only few participants; some only one participant. Therefore, including examiners as a factor in the ANOVA was not feasible. Numbers of participants per each examiner are included in S6 Table. Future attempts to adapt methodologies using non-laboratory equipment should emphasize standardizing the environment in which measurements take place to control environmental stimuli. As the results showed, subjects' expectations correlated with the pain they experienced. Also, test subjects were able to predict the intensity of pain. The present study is limited to making inferences at the association level, rather than establishing causation. Further research should focus on experimentally manipulated expectations to test if subjects' expectations affect SSp. This could give an insight on the causal relationship between expectancy and SSp.

## Conclusions

This study successfully replicated the SSp effect using CPT. It proposes a new method to induce SSp using a non-laboratory setup. Also, three novel aspects emerged from this study. Firstly, SSp can be assessed outside of the laboratory, providing a new tool for experiments outside of the laboratory in e.g., clinical settings. Secondly, it seems that SSp can be shaped by a mixture of excitatory and inhibitory mechanisms and is influenced by the temporal summation of the nociceptive system. Also, spatial summation seems to be linked with expectations, but future studies that directly modulate expectations are needed to investigate their influence on SSp. Lastly, the inclusion of physiological measures could also be beneficial in future studies.

## Supporting information

**S1 Text. Protocol deviations.**
(DOCX)

**S2 Text. Exclusion criteria.**
(DOCX)

**S1 Fig. Study procedures. Sequence of conditions was random, "ascending" is presented as an example.** Study procedures. In each condition single trial lasted 60 seconds (regardless of the number of segments involved). Each trial started with a question about expected pain intensity. Before and after each experimental condition cold pain thresholds ($PT_{COLD}$) were tested on the examined limb. Participants were instructed to immerse their hand up to the line which separated a given number of segments. Participants were prompted to rate their pain intensity on the VAS scale at the following time points: after 10, 30 and 50s. Inter-trial intervals were set at 5 minutes. The interval between each condition was one hour.
(TIF)

**S1 Table. Means and standard deviations for temperatures in pilot study (training).** SD-standard deviations.
(DOCX)

**S2 Table. Measurement of pain thresholds.** SD- standard deviations.
(DOCX)

**S3 Table. Means and standard deviations for pain intensity reported during cold water immersions.** 1/5 –Segment 1, 2/5 –Segments 1 to 2, 3/5- Segments 1 to 3, 4/5 –Segments 1 to

4, 5/5- Segments 1 to 5, SD, standard deviations.
(DOCX)

**S4 Table. Slopes and intercepts of the relationships between pain and number of stimulated segments.** 10s - pain intensity measured after 10 seconds, 30s - after 30 seconds, 50s - after 50 seconds of immersion, Mean - mean pain intensity from each immersion, B - unstandardized coefficients, SE - standard error, β - standardized coefficients, *p*–significance value.
(DOCX)

**S5 Table. Pain-related expectations measured prior to cold water immersions.** 1/5 –Segment 1, 2/5 –Segments 1 to 2, 3/5- Segments 1 to 3, 4/5 –Segments 1 to 4, 5/5- Segments 1 to 5, SD, standard deviations. M, median. IQR, interquartile range.
(DOCX)

**S6 Table. The number of participants tested by each examiner.**
(DOCX)

**S1 File. Raw data.**
(TXT)

## Acknowledgments

The authors would like to thank Magdalena Grygiel and Daniel Adamczyk for help with conducting the experiment.

## Author Contributions

**Conceptualization:** Jakub Nastaj, Jacek Skalski, Aleksandra Budzisz, Tibor M. Szikszay, Sylwia Swoboda, Weronika Kowalska, Daria Nowak, Edyta Zbroja, Natalia Kruszyna, Marta Jakubińska, Dominika Grygny, Andrzej Małecki, Kerstin Luedtke, Wacław M. Adamczyk.

**Data curation:** Jakub Nastaj, Romuald Polczyk.

**Formal analysis:** Tibor M. Szikszay, Romuald Polczyk, Wacław M. Adamczyk.

**Investigation:** Jakub Nastaj, Jacek Skalski, Aleksandra Budzisz, Sylwia Swoboda, Weronika Kowalska, Daria Nowak, Edyta Zbroja, Natalia Kruszyna, Marta Jakubińska, Dominika Grygny.

**Methodology:** Jakub Nastaj, Jacek Skalski, Tibor M. Szikszay, Sylwia Swoboda, Weronika Kowalska, Daria Nowak, Edyta Zbroja, Natalia Kruszyna, Marta Jakubińska, Dominika Grygny, Kerstin Luedtke.

**Project administration:** Jakub Nastaj, Jacek Skalski, Andrzej Małecki, Wacław M. Adamczyk.

**Resources:** Jakub Nastaj.

**Supervision:** Jakub Nastaj, Kerstin Luedtke, Wacław M. Adamczyk.

**Validation:** Romuald Polczyk.

**Visualization:** Aleksandra Budzisz.

**Writing – original draft:** Jakub Nastaj, Jacek Skalski, Aleksandra Budzisz, Tibor M. Szikszay, Sylwia Swoboda, Weronika Kowalska, Daria Nowak, Edyta Zbroja, Natalia Kruszyna, Marta Jakubińska, Dominika Grygny.

**Writing – review & editing:** Jakub Nastaj, Romuald Polczyk, Andrzej Małecki, Kerstin Luedtke, Wacław M. Adamczyk.

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
