## [Decision Letter · Decision Letter 0]

25 Jun 2023

PONE-D-23-02600Expectancy effects in spatial summation of pain using a home-based paradigmPLOS ONE

Dear Dr. Adamczyk,

Thank you for submitting your manuscript to PLOS ONE. After careful consideration, we feel that it has merit but does not fully meet PLOS ONE’s publication criteria as it currently stands. Therefore, we invite you to submit a revised version of the manuscript that addresses the points raised during the review process.

We look forward to receiving your revised manuscript.

Kind regards,

Alessandro Vittori, M.D.

Academic Editor

PLOS ONE

“The authors declare no conflict of interest. The authors would like to thank Magdalena Grygiel and Daniel Adamczyk for help with conducting the experiment. JN and JS and WA are supported by the National Science Center within the grant no. 2020/37/B/HS6/04196.”

3. We noted in your submission details that a portion of your manuscript may have been presented or published elsewhere. [DETAILS AS NEEDED] Please clarify whether this [conference proceeding or publication] was peer-reviewed and formally published. If this work was previously peer-reviewed and published, in the cover letter please provide the reason that this work does not constitute dual publication and should be included in the current manuscript.

Additional Editor Comments:

Unfortunately the manuscript is not acceptable in its present form. Please follow the suggestions of the reviewers to re-evaluate publication.

Kind Regards

Reviewers' comments:

Reviewer's Responses to Questions

**Comments to the Author**

1. Is the manuscript technically sound, and do the data support the conclusions?

Reviewer #1: Yes

Reviewer #2: Partly

Reviewer #3: Yes

2. Has the statistical analysis been performed appropriately and rigorously? 

Reviewer #1: Yes

Reviewer #2: Yes

Reviewer #3: Yes

3. Have the authors made all data underlying the findings in their manuscript fully available?

Reviewer #1: Yes

Reviewer #2: Yes

Reviewer #3: Yes

4. Is the manuscript presented in an intelligible fashion and written in standard English?

Reviewer #1: Yes

Reviewer #2: Yes

Reviewer #3: Yes

5. Review Comments to the Author

Reviewer #1: General comments

This study is well done and the general message is very clear (as it is important and relevant). There are a few things that could improve this manuscript in my opinion. I've tried to give suggestions how to do so. The most important suggestion is to 'allign' all the segments; the order of the aims in the intro/methods/results/discussion/conclusion is inconsistent and a overview of aims is lacking, as such, a general overview is missing and the paper is harder to read. An overview of aims in the order of importance (which I think is done in the conclusion) at the end of the introduction would greatly help (next to staying consistent in this chosen order).

Further, more attention to the uniqueness of this design (home vs lab, multiple environments/examiners vs 1 lab) would increase the value of this paper and assist in further research using this type of (pragmatic) design.

Since the study is done very well (including the analyses and reporting of results), a more clear and consistent order and overview of aims would greatly benefit the manuscript and will allow readers to have a better understanding of the message this study delivers.

Abstract

The first sentence overvalues the COVID-19 part and underestimates the value of the paper due to this. If COVID needs to be named, then surely not in the first AND last sentence (which are powerpositions). The message of this paper is (in my opinion) way more far reaching then this. For me, the COVID sentences can actually be skipped fully.

Introduction

General: The order/structure is somewhat confusing, since the aims (sometimes called "move 3" in the 'Creating a Research Space' method of writing, https://libguides.usc.edu/writingguide/CARS) are presented before the reasoning. Consequently, the multiple aims are scattered and an overview (typically a the end) of the aims is missing (see comment later)

r49 - is 'irritated' the correct word? Immersed would be sufficient, since the actual activating of nociception is explained directly after.

r56-58 - The addition of this reasoning does not seem to introduce something which is tested in this paper. If it is for discussion purposes, i'd suggets moving it to the discussion. Its now a 'stand alone'.

r59-61 - Two aims are posted. I'd suggest rewriting the first sentence to show why replication is of importance (and in the end, show in the aims that replication is one of them). The second part (r60 and on) introduces a new aim, while its not supported yet. First introduce expectation and why it is important to test..and move the aim to the end.

r63 - A bit more is known about expectancy effects. Based on the scope of the paper, i'd suggest referring to the expectancy paradigms available and according references. CPM / Placebo is a bit too 'superficial' for this paper in my opinion, specifically when its an aim.

r64 - It seems a new aim is now introduced, eventhough its still the expectation part. I'd suggest to rephrase this as connecting expectation and participants' predictions, but not as a aim here.

The last paragraph of the introduction then can show the overview of the aims, since there are multiple;

Reproduction of the adopted methodology (with addition of the aim of the methodlogy)

Expectancy vs effect

Non-laboratory setting (which should be written carefully, since the current setting can not be compared to a lab setting directly).

Also, consider the order of these aims in relation to the analysis. Further, the order of aims should be consistent with the order of the rest of the introduction (move 1 / move 2), the results order etc. An overview of the aims will aid the reader to have a better understanding, since multiple aims are covered (and they all are important in my opinioin).

Last aim; this aim is more important then covid - if this design could be done, it enables research groups with less outiled labs (e.g. Universities of Applied Science in the EU) the reproduce or conduct preliminary studies. I'd suggest to enrich this to increase its importance. Covid "inspired" and forced to think differently, but the consequenes may be way more fruitfull.

Methods

General: Clear and reproducable methods, which logically allign with the aims.

Healthy participants - possibly add the reference of healthy participants in QST studies (https://pubmed.ncbi.nlm.nih.gov/26313407/). The amount of participants completing the study would be a result i'd say? As such, it's now doubled in both the method and result. Only in the results is enough.

There is not much information regarding the training of the examiners. Since there are 13, which efforts were made to obtain equal methods used during measurements? What does a 'intensive training' mean? Further, is there any quantitive information available about how 'good' they measure (for example, intra-examiner values which may be compared)? If not (which I assume), is there any qualitative info available; how was confirmed it was 'good enough' to be precise? Since this study is particularly unique in the 13 different settings/examiners, it would help to add a bit of info (if available) on this matter. That would also help to inspire future research using these kinds of designs.

Further, there is no information about the setting in terms of environment; where was it done in the non-laboratory setting / household; in the living rooms, in an available room? A bit more information would help. Think in terms of temperature, noise, static distractions due to paintings etc.

I can imagine the reader misses a familiarisation phase, which seems to be very necessary in QST based research. However, due to the aim of including expectation, this is logically not within the design. I'd suggest to think adding a short sentence to adress the purposefully lacking familiarisation for the aim of the study, in order to allign with this matter in QST studies. However, if the authors are convinced it's better without, it's okay as well.

-Statistics

Statistics are done logically and are reproducable - they are adequately chosen for the research aim (and specifically also for the further effects, which would not be doable with oversimplified statistics).

Since many different 'settings - examiners' pairs were used (as in, each examiner had its own setting), is the factor reflecting this concept of any relevance? Multiple studies show effect of environment, instruction, therapeutic alliance and even the examiner self have an influence. Eventhough its not disentangleble, it could be tested within the model (and possibly the data is actually clustered then, with the setting as cluster, and a Linear Mixed Model could/should be used to add the hierarchy of clustering - eventhough I'd understand the primary analises with assumption of non-clustering is focused differently). Evenmore, possibly this is already tested and not reported to keep overview - but i'd advise to look in to it due to the uniqueness of this study in these different settings. A short statement of its effect would inform the reader in regards to the unique design.

Results

Clear and full reporting of the results, including interaction effects and their meaning.

I miss information about the clustering of participants over examiner (how many per examiner?). Following the 'statistics' comments - it would be of added benefit to add a quantification of the effect of examiner/setting, if the author group agrees. If not, just a bit of info on the clustering would be enough.

Table 1. I see no added benefit of 2 decimal places in the scores - one is enough to maintain precision, but will increase overview and interpretation (e.g. less distraction). Generally, I'd suggest adding only one decimal if the raw measurement was without decimals, as these presented constructs are. For the catagorical, add % for quick interpretation.

Table 3. The addition of * p < .05 is not usefull, since all are <.001 and ***.

r319 - eventhough it is interesting, the use of the word 'interestingly' suggest a contrast to the sentence before and is as such a bit confusing for me. Next / Further would be more pure for me.

Discussion

Generally: The order of the presented discussion is not consistent with the order of the aims/intro/methods/results -> Here, the expectation is the primary aim, in the intro the reproduction is the first presented aim, in the methods and results, it's the effect itself (segement vs Ssp effect, then followed by a time effect).

r 345; the order of the aims written here are not consistent with the introduction, methods and results.

r 366; 'significant in 9/10 cases' -> in the result (table III), all cases are highly significant?

r429; to consider (as commented in the introduction as well): I don't think its about spatial summation during pandemic - its about spatial summation outside of the lab what is the most important message here (which is discussed in this paragraph as well, as it should). I truly hope we will not have such a pandemic anymore, but I do hope with this innovative design we can do robust research in non-lab settings (which is also necessary sometimes for more translational studies). I'd suggest rephrasing the heading to "Spatial summation outside a laboratory environment" (or something like that).

Paragraph r429 - I miss the discussion specifically about the home environment and differences between the setting/examiners (as commented on earlier). Future research could greatly benefit on this methodology, but it has to be discussed specifically. As it is now, only the 'why' and the method (used in all settings) is discussed (as it is adapted to less expensive equipment), but not the challenges that could be encountered. Please add a deeper discussion about the home environment AND the different settings used due to having multiple examiners/environments.

Reviewer #2: According to the public registration, the study of Nastaj and colleagues aimed to answer the following questions :

-Can spatial summation of pain be observed in a home-based environment?

-Are pain expectations related to the perceived pain intensity during the spatial summation paradigm?

-Does the temperature of the water used for the spatial summation paradigm confound the spatial summation effect?

Even though it is not clearly stated as a hypothesis, the authors also aimed at assessing the replicability of the order effect observed by Marchand and Arsenault.

I think that the data presented in the manuscript can answer all these questions.

However, I have some general reservations on the interpretation that is made of some of the observed effects and more specific comments on parts of the manuscript. Also, I think that the manuscript would benefit from an extra round of proof-reading as some typos are still present.

MAIN POINTS:

Expectations

I think there is some confusion on the treatment of expectations and their effects on pain in the manuscript.

The Authors show that the reports of expected and perceived pain are correlated. In my opinion, this shows that the participants have a good internal model of their nociceptive system and of the effect of stimulation surface area on the intensity of their pain sensation (spatial summation). This shows that participants expect larger stimuli to be more painful (in a way, an effect of spatial summation on expectations).

This, however, does not show an effect of expectations on spatial summation. To show that the spatial summation effect is modulated by expectation, the Authors should have manipulated the expectations of the participant and observed a change in perceived intensity congruent with the expectation manipulation, similarly to the studies that the authors cite in the introduction and discussion sections. Citations of these studies in conjunction with ambiguous phrasing of the effects observed by the authors in their own data makes the discussion slightly misleading.

Sensitization

For each immersion, perceived pain was reported three times: after 10, 30 and 50 s. Pain ratings are shown to increase between these time points. The Authors interpret this effect of time as sensitization. I don’t think that sensitization is the most likely explanation for this effect.

I am almost certain that (at least some of) this effect can be explained by continuously decreasing temperature at the depth of the afferents. The temperature at the depth of the nociceptors is not the same as the temperature at the surface of the skin, it is a compromise between the temperature of the surface of the skin and the deep body temperature. Following a change of the surface temperature, the temperature at the depth of the afferents will (slowly, as heat needs to be conducted through tissues largely made of water) drift towards a new equilibrium. Given that cold nociceptors are thought to be located deeper in the skin than most nociceptors and given the large temperature delta caused by the immersion, the temperature at the depth of these nociceptors can be expected to significantly differ at 10, 30 and 50 s post immersion.

Additionally, temporal summation can be expected to play a role to the effect of time on pain intensity.

Descending pain inhibition

The order of presentation of the different stimuli (increasing or decreasing sequence) influenced the reported level of pain, at least for the smallest surface of stimulation. The Authors interpret this as evidence for a modulation of the spatial summation effect by descending inhibitory mechanisms.

Even though I don’t think that it is completely unlikely, I also don’t find the evidence compelling and I think that the Authors reach this conclusion without sufficient argumentation. I would therefore suggest that they revise the discussion to reflect the thought process that led them to this conclusion.

Additionally, I think that the Authors should also consider the possibility that this order effect does not reflect a true difference in pain but is rather an artefact of measurement. Previous stimuli/perceptions can influence the way the participants use a rating scale.

For this reason, evaluation of spatial summation effects should actually not be done with an ordered sequence such as the one used in this manuscript but rather with a random ordering of the different surfaces. Of course, as the authors aimed at replicating the order effect, using an ordered sequence made sense for their study.

SPECIFIC POINTS

-“Spatial summation of pain (SSp) can be defined as an increase of pain intensity when body area (1–3) or distance (4,5) between stimulated areas are increased (3).” (41-42)

The definition of spatial summation put forward by the authors is not the conventional definition. It misses the fact that spatial summation affects not only the perceived intensity of suprathreshold stimuli but also the threshold. Additionnaly, the distance between two stimuli modulates spatial summation but is not spatial summation per se and the relationship between stimulus distance and perceived intensity is not as simple as described here (at some point the effect saturates or even reverse).

-“[…] is that there is no linear increase in pain intensity during a linear increase in the stimulated body area (1) or in the distance between the stimuli (1,4), further suggesting that inhibitory processes are involved in SSp.” (45-47)

I am not sure I understand why the authors think the non-linearity of the stimulus surface - perceived intensity suggests that inhibitory processes are involved. Most stimulus intensity – perception functions are non-linear. Additionally, the non-linearity of this relationship is not obvious based on the data presented in this manuscript.

-“The low temperature activates nociceptive fibers (13,17,18) through low temperature sensitive ion channels TRPM8” (49-50)

The identity of the noxious cold receptors remains a matter of debate. Aside from TRPM8, TRPA1 is also considered as a good candidate and there is evidence suggesting that a third channel could be involved.

-“CPT is a suitable method to study SSp; the stimulated bodily area can be adjusted in a controlled and standardized manner by immersion depth (2,15).” (51-52)

I agree that CPT can be used to study spatial summation. However, I think that this sentence minimizes the fact that this approach does not actually allow to control the exact stimulation surface area, as every participant’s anatomy is different.

-“The size of areas exposed to cold stimulation is presented in Figure 2.” (138-139)

It is not clear to me how this size was measured. I don’t think it is currently explained in the manuscript. I think it is important to describe it.

-“In the first stage, the effect of stimulation area on pain intensity was investigated using a General Estimated Equations (GEE) model with three within subject factors: “condition” (ascending, descending), “segment” (1, 2, 3, 4, 5 segments) “time” (10s, 30s, 50s).” (187-189)

Where the different levels of factors “condition”, segment”, and “time” treated as factors or continuous variables?

In the supplementary materials, the authors indicate that they decided to use a GEE instead of a GLM (as per registration) because the pain ratings were not normally distributed. Non normality of the outcome variable is not a violation of the assumptions of the GLM. Non normality of the residuals (the difference between the values predicted by the models and the true data) is. If the authors did, indeed, not try fitting a GLM because the pain ratings were not normally distributed, I think they should fit such a model and then assess the normality of the residuals. If those are indeed normally distributed, the authors should report the results of the GLM rather than the GEE. That way, they would deviate less from their registration.

Similarly, normality of the data is not a condition to use Pearson’s correlation and I think the Authors should stick to the metric they initially chose. As the existence of a linear relationship is a condition to use this metric, I would recommend including scatter plots of the data along the correlation coefficients, so that the reader can verify that the relation is approximate linear.

In parallel to what I mentioned in the previous paragraph, it is good practice to include visualization of the raw data and not just the marginal means/CI in the figures.

-“Mean temperature of the cold water was 5.12°C (��0.47) during the ascending condition (t[66] = -0.5, p = 0.63) as well as during the descending condition (t[66] =1.2, p = 0.22) no significant differences (before condition vs. after condition) were shown for the pain thresholds, indicating that pain sensitivity was stable over the course of experiment (Appendix 4).” (225-229)

I don’t think that these analyses are described in the method section. I think they should.

- “Descriptive statistics for expectation are presented in Figure 3 and Table 2.” (284)

Are the plotted values marginal means or sample means? The text sometimes indicates one think and sometimes the opposite.

-“No difference between expectancy measured before the first immersion (unbiased) versus expectancy measured on a trial-by-trial basis were found (Wald �2(1) = 1.52, p = 0.22), thus it was decided to perform main expectancy analysis on the pooled dataset.” (284-287)

I don’t think that this analysis has been described in the method section. I think it should.

Additionally, I don’t think that the absence of a statistical difference is a good reason to pool the data and I would like the before trial and during trial expectancy data to be analyzed separately, as pooling may obscure an effect of the previous stimulus on expected for the next stimulus.

-“Interestingly, the repeated measures ANOVA of the size of immersed segments showed a significant effect for “size”, indicating that the area of stimulation increased from trial to trial (F(1,67) = 135.17, p < 0.001, �2 p = 0.70).” (320-321)

I don’t think that this analysis has been described in the method section. I think it should.

Reviewer #3: The present study investigates new aspects of SSp by exploring the involvement of expectations. At the same time, this study replicates previous results, slightly changing the experimental design; given the replication crisis, replicating previous data must be valued. An additional novelty of this study is the introduction of a new feasible way to induce SSp using a home-based set-up. The experiment is methodologically sound and leads to novel findings, including the association between expectations and perceived pain in SSp. The manuscript is generally well-written and clear. The discussion could benefit from revision and reorganisation, empathising which results are the most novel and which ones are not. Some additional details should be added in the analysis and in the results section. Overall, this is an interesting and thoroughly designed work.

Abstract.

Line 30-33. These two novel findings should be highlighted in the discussion. Here it is clear that these two findings are a novelty, while in the discussion (i) is not presented as a novelty. In the abstract, I suggest mentioning novelty ii- expectation) first, and then the other novelty (I - spatial summation and time), because the investigation of expectation is one of the primary aims of the study.

Line 32- 33. ‘indicating that pain expectations can contribute to SSp’. Either replace ‘can’ with ‘could’ or rephrase the sentence with something like: indicating a link between expectations and SSp.

Design and method.

Please specify whether the examiners physically went to the participants’ houses or whether the participants physically went to the examiners’ houses. Please clarify this point also in relation to the covid situation (was it during a time in which it was possible to go to each other houses but not in the lab?). Or maybe I misunderstood, and it was done online? Please clarify this point.

113 Remove comma after ‘fact’

Figure 1. Since there is a drawing for the ascending session, I suggest adding the drawing for the descending one (although this is not necessary).

Just by reading the Experimental Design section, it is not clear whether the immersion of each hand section is sequential with breaks between each segment test, or whether each segment test occurs one after the other, without breaks. The experiment design becomes clearer when reading the ‘Trial Design’ section in which this information is specified. However, it would be best if this structure was clearer from earlier on in the manuscript. I suggest adding a few sentences that clarify this also in the Experimental Design section, while further details are then given in the ‘Trial Design’. Please also add a reference that backs up the choice of using a 5 minutes break between each trial as a sufficient time to reach the baseline temperature level.

158 ‘Before and after each experimental condition, the cold pressor threshold was tested’. Was this done before and after each segment? Or just before and after the ascending and the descending. Please clarify. If the cold pressor threshold was not measured before each trial, this could be mentioned as a potential limit.

182 I would replace ‘conduct’ with ‘run’

Results.

226 was the temperature identical for the descending and the ascending? 5.12°C

Clarification is required, both in the statistical analysis (line 244) and in the results section, regarding the Bonferroni-corrected p-value. Please specify if the p-values reported in the manuscript are already corrected. For each interaction, please also specify the number of planned comparisons and how this was used to correct for Bonferroni. My understanding is that for each interaction there are 5 comparisons (comparison 1: 1/5 ascendent vs 1/5 ascendent - comparison 2: 2/5 ascendent vs 2/5 ascendent - comparison 3: 3/5 ascendent vs 3/5 ascendent - comparison 4: 4/5 ascendent vs 4/5 ascendent - comparison 5: 5/5 ascendent vs 5/5 ascendent), if this is true the corrected p-value threshold should be 0.01. If the corrected p-value is 0.01, some of the significance of some results should be rediscussed (es. p=0.017 for the ‘condition’ x ‘area’ x ‘time would be almost significant. Given that Bonferroni is very strict, this should not be a problem but should still be acknowledged). However, I might be missing something here, and the p-values might have already been corrected (i.e., multiplying the p-value by 5). Please, clarify this aspect for all the interaction effects (es. ‘Condition’ x ‘segment’; ‘segment’ x ‘time’; ‘condition’ x ‘area’ x ‘time’.

Line 130-131. The authors state that anger measurements have been taken. However, these data are not included in the analysis. Please, explain why this interesting variable has not been considered. If no association between fear of pain and pain has been found, I would still recommend mentioning this as an insignificant result and adding a brief comment on why this might be the case (e.g. would be interesting to check whether fear of pain is a predictor of perceived intensity at the first trial. Individuals with higher fear of pain might experience greater pain in the first pain test. The first trial could be interesting because, given the novelty of the experience, this might be more influenced by fear).

Discussion.

Line 353 – 354 the authors state ‘that spatial summation i) can be inferred from subjects’ pain expectations and ii) is strongly influenced by sensitization, iii) is likely shaped by descending pain inhibition.’ While these are important points, the link between each point and the findings of this study are not always clear throughout the discussion. I suggest making it very clear in the discussion how the results of this study support each of these claims.

Line 387. Before 'Despite' need full stop.

394 immersion of segment 1 was slightly ‘higher’? Maybe I am missing something, but my understanding is that pain perception of segment 1 in the descending condition is ‘lower’.

In the section ‘Physiological mechanisms of SSp’ the distinction between facilitatory and inhibitory processes and how these are linked to your results could be clearer. My understanding is that from line 393 to line 399 the authors are talking about inhibitory mechanisms, while from line 399 to line 407 the authors discuss facilitatory responses. Please, make this distinction clearer.

A limitation and future directions section could be added.

Conclusions.

Line 456. Please keep the hierarchy of the aims fixed across the manuscript. At lines 345 -348 the aims are well stated, I recommend keeping this hierarchy throughput the text (i.e., also in the abstract and in the introduction, in which the order is, at times, mixed). Since one of the main aims was to replicate the previous findings with a different design, I suggest mentioning this in the conclusion.

Line 457 ‘were emerged’: remove ‘were’

Line 457 – 459. In the present study expectations were shown to predict pain perception, however, these were not modulated and therefore no conclusion can be drawn on the potential effects of expectations on pain. I suggest writing something on the lines of: Firstly, spatial summation seems to be strictly linked with expectations, but future studies that directly modulate expectations are needed to investigate their influence on SSp, and physiological measurements should also be measured.

Line 459 ‘confirmed’ becomes ‘confirm’

6. PLOS authors have the option to publish the peer review history of their article (what does this mean?). If published, this will include your full peer review and any attached files.

Reviewer #1: **Yes: **Roland Reezigt

Reviewer #2: No

Reviewer #3: **Yes: **Eleonora Maria Camerone

---

## [Author Response · Author response to Decision Letter 0]

19 Aug 2023

Reviewer #1

This study is well done and the general message is very clear (as it is important and relevant). There are a few things that could improve this manuscript in my opinion. I've tried to give suggestions how to do so. The most important suggestion is to 'allign' all the segments; the order of the aims in the intro/methods/results/discussion/conclusion is inconsistent and a overview of aims is lacking, as such, a general overview is missing and the paper is harder to read. An overview of aims in the order of importance (which I think is done in the conclusion) at the end of the introduction would greatly help (next to staying consistent in this chosen order).

Response: Thank you very much for your suggestions. We rearranged the manuscript to allow a better flow of the sections intro/methods/results/discussion/conclusion and to enhance the comprehensibility of the manuscript. 

Further, more attention to the uniqueness of this design (home vs lab, multiple environments/examiners vs 1 lab) would increase the value of this paper and assist in further research using this type of (pragmatic) design.

Response: We would like to thank the reviewer for the valuable comments. In the revised version of our manuscript, we paid more attention to describe the new methodology in a non-laboratory environment (see specific changes made below).

Since the study is done very well (including the analyses and reporting of results), a more clear and consistent order and overview of aims would greatly benefit the manuscript and will allow readers to have a better understanding of the message this study delivers.

Response: Thank you very much for your feedback and suggestions. The results section has been restructured in order to allow a better understanding of the results.

Abstract

The first sentence overvalues the COVID-19 part and underestimates the value of the paper due to this. If COVID needs to be named, then surely not in the first AND last sentence (which are power positions). The message of this paper is (in my opinion) way more far reaching then this. For me, the COVID sentences can actually be skipped fully.

Response: Thank you for your feedback. The first sentence of the abstract was removed. 

The last sentence was modified to: “Moreover, this study proposes a new feasible way to induce SSp using a home-based set-up”

Introduction

General: The order/structure is somewhat confusing, since the aims (sometimes called "move 3" in the 'Creating a Research Space' method of writing, https://libguides.usc.edu/writingguide/CARS) are presented before the reasoning. Consequently, the multiple aims are scattered and an overview (typically at the end) of the aims is missing (see comment later)

Response: The order of the introduction section was corrected and rewritten. Please review changes made.

r49 - is 'irritated' the correct word? Immersed would be sufficient, since the actual activating of nociception is explained directly after.

Response: The word “irritated” was changed to “immersed”.

r56-58 - The addition of this reasoning does not seem to introduce something which is tested in this paper. If it is for discussion purposes, i'd suggets moving it to the discussion. Its now a 'stand alone'.

Response: That sentence “The authors proposed an explanation that in the increasing condition, both facilitatory and inhibitory mechanisms were gradually being recruited at the same time, thus they were interfering with each other (Marchand & Arsenault, 2002)”was moved to the discussion subsection “Reproducibility of spatial-related effects”.

r59-61 - Two aims are posted. I'd suggest rewriting the first sentence to show why replication is of importance (and in the end, show in the aims that replication is one of them). The second part (r60 and on) introduces a new aim, while its not supported yet. First introduce expectation and why it is important to test and move the aim to the end.

Response: Thank for this comment. The introduction section was reorganized. Now, importance of replication and investigation of the associations with expectations are introduced separately and specific aims are moved to the end of introduction section.

r63 - A bit more is known about expectancy effects. Based on the scope of the paper, i'd suggest referring to the expectancy paradigms available and according references. CPM / Placebo is a bit too 'superficial' for this paper in my opinion, specifically when its an aim.

Response: The section about expectations was rewritten within the introduction. Now, we refer to the Predictive Coding and Bayesian integration framework (e.g., see Song et al., 2021; Büchel & Geuter, 2014). Please see introduced changes here:

“The role of expected (predicted) value on pain perception could be explained using the predictive coding framework (Song et al., 2021; Büchel & Geuter, 2014), wherein the prior distribution of predicted values leads to the shift of posterior distribution following the integration with the collected data, here nociceptive information.”

r64 - It seems a new aim is now introduced, even though it’s still the expectation part. I'd suggest to rephrase this as connecting expectation and participants' predictions, but not as a aim here.

Response: Thank you for this comment. Two sentences about the aim of investigating expectations were rephrase as one: 

“Furthermore, the current experiment aimed to investigate the potential association between pain-related expectations and SSp by examining participants’ predictions regarding pain using an SSp paradigm” 

The last paragraph of the introduction then can show the overview of the aims, since there are multiple;

Reproduction of the adopted methodology (with addition of the aim of the methodology)

Expectancy vs effect

Non-laboratory setting (which should be written carefully, since the current setting cannot be compared to a lab setting directly).

Also, consider the order of these aims in relation to the analysis. Further, the order of aims should be consistent with the order of the rest of the introduction (move 1 / move 2), the results order etc. An overview of the aims will aid the reader to have a better understanding, since multiple aims are covered (and they all are important in my opinioin).

Response: Thank you for your feedback. The order of the introduction section was corrected and rewritten. The structure of the introduction is now as following: First, the characteristics of spatial summation of pain and the cold pressor task paradigm. Then, how expectations can modulate pain and the role and importance of performing replication studies. The introduction concludes by outlining the aims of the study in the following order: to perform replication of Marchand & Arsenault study, to apply the new methodology outside the laboratory setting, and to investigate if there is a link between expectations and spatial summation of pain.

Last aim; this aim is more important then covid - if this design could be done, it enables research groups with less outiled labs (e.g. Universities of Applied Science in the EU) the reproduce or conduct preliminary studies. I'd suggest to enrich this to increase its importance. Covid "inspired" and forced to think differently, but the consequenes may be way more fruitfull.

Response: We are very thankful for this comment. We rephrased the section in the introduction related to the new methodology. 

Please see introduced changes here: 

“Based on previous studies on the CPT paradigm using non-laboratory equipment without water circulation (Uman et al., 2006 ;Vigil et al., 2014), the authors decided to additionally test if it is possible to conduct and replicate, for a first time, SSp outside the laboratory setting (Skalski et al., 2021). This novel yet adopted methodology could enable the conducting of similar replications and preliminary studies for researchers who do not have dedicated equipment for CPT in their laboratories, or it could be used in exceptional situations when conducting research in laboratories is hampered (Miki et al., 2020; Sohrabi et al., 2021). Moreover, the proposed method could be very useful for clinical purposes if there is lack of dedicated CPT equipment”.

Methods

General: Clear and reproducable methods, which logically allign with the aims.

Response: We are very thankful for your positive feedback.

Healthy participants - possibly add the reference of healthy participants in QST studies (https://pubmed.ncbi.nlm.nih.gov/26313407/). The amount of participants completing the study would be a result i'd say? As such, it's now doubled in both the method and result. Only in the results is enough.

Response: The information about the amount of participants completing the study was moved to the results section.

A reference was added to the “study population section” before listing exclusion criteria:

“A thorough interview was conducted using a screening questionnaire which allowed to enroll healthy participants (Coghill & Yarnitsky, 2015 )”

There is not much information regarding the training of the examiners. Since there are 13, which efforts were made to obtain equal methods used during measurements? What does a 'intensive training' mean? Further, is there any quantitive information available about how 'good' they measure (for example, intra-examiner values which may be compared)? If not (which I assume), is there any qualitative info available; how was confirmed it was 'good enough' to be precise? Since this study is particularly unique in the 13 different settings/examiners, it would help to add a bit of info (if available) on this matter. That would also help to inspire future research using these kinds of designs.

Response: Thank you for this thoughtful comment. We agree that information about examiners training could be beneficial for readers. A section about training the examiners was added to the “general information”: 

“The home setting differed between examiners. Settings were chosen such that the subject could stay without distractions such as music or the sounds of other household members. As the study was conducted during the COVID-19 pandemic, study participants were family members of the examiners, people they lived with or other relatives who were not covered by government restrictions. Examiners followed an intensive training for the precise data collection and screening procedures. Each examiner was involved in all phases of study preparation. Firstly, they took part in a series of online meetings during which they were introduced to the design of the experiment. Secondly, they received training on the preparation of the CPT (see below). During training, water temperatures achieved in their home environment were measured (water temperature in CPT before adding 6 foil ice-cube packs and water temperature after adding ice cubes every 10 minutes for 2h). Thirdly, examiners conducted a pilot study, which was preceded by training in the laboratory. Each of the examiners was individually trained how to follow each step of the study SOP (standard operating procedure) and was supervised by JN. Each examiner conducted a pilot assessment of one single individual in their home environment. To investigate if the CPT method was consistent among examiners, measures of temperatures were compared (see S1 Table). Finally, before conducting main experiment, examiners were demonstrating all procedures to JN, who assessed them in terms of quality and ensured feedback if any violations from SOP were detected.” 

Further, there is no information about the setting in terms of environment; where was it done in the non-laboratory setting / household; in the living rooms, in an available room? A bit more information would help. Think in terms of temperature, noise, static distractions due to paintings etc.

Response: We agree that detailed information about household settings would be very useful. Unfortunately, we did not gather any data about household setting apart from temperatures which we did believe were one of the most important. Examiners had to follow instruction about choice of setting. We added information about this in “General information” section:

“The home setting differed between examiners. Mainly living rooms, bedrooms and kitchens were chosen to build up the setup. Settings were chosen such that the subject could stay without distractions such as music or the sounds of other household members.” 

I can imagine the reader misses a familiarisation phase, which seems to be very necessary in QST based research. However, due to the aim of including expectation, this is logically not within the design. I'd suggest to think adding a short sentence to adress the purposefully lacking familiarisation for the aim of the study, in order to allign with this matter in QST studies. However, if the authors are convinced it's better without, it's okay as well.

Response: Thank you for this comment. Indeed, we did not include a familiarization phase to not induce any expectations of pain from cold temperature before the main phase of the experiment. However, to address this lack of familiarization, we added a sentence in the “experimental design” section:

“A familiarization phase was not embedded within the study design to not bias expectations regarding pain from cold water immersion. However, measuring cold pain thresholds (PTCOLD) prior to the main data collection, allowed participants to familiarize with the cold sensations without inducing expectations.”

Statistics

Statistics are done logically and are reproducible - they are adequately chosen for the research aim (and specifically also for the 

further effects, which would not be doable with oversimplified statistics).

Response: Thank you very much for your positive feedback. We also think that GEE model was more accurate in the case of our data. In fact, both, measured outcome, and their residuals were not in line with normal distribution (see our response to reviewer #2). Please note that any protocol violation / change has been transparently reported in the manuscript/supplements.

Since many different 'settings - examiners' pairs were used (as in, each examiner had its own setting), is the factor reflecting this concept of any relevance? Multiple studies show effect of environment, instruction, therapeutic alliance and even the examiner self have an influence. Eventhough its not disentangleble, it could be tested within the model (and possibly the data is actually clustered then, with the setting as cluster, and a Linear Mixed Model could/should be used to add the hierarchy of clustering - eventhough I'd understand the primary analises with assumption of non-clustering is focused differently). Evenmore, possibly this is already tested and not reported to keep overview - but i'd advise to look in to it due to the uniqueness of this study in these different settings. A short statement of its effect would inform the reader in regards to the unique design.

Response: We are very thankful for this comment. Indeed, the environment and especially context seem to play a role in pain experiments (see some seminal examples: Carlino & Benedetti, 2016; Colloca et al., 2004; Moseley & Arntz, 2007). Given the circumstances and the design of the current study, it was not intended to test the effects of context/environment. We would thus prefer to keep this analysis out of the scope of the current investigation. However, we do agree that the variety of settings could -to some degree- confound the results.

Unfortunately, a direct test of the influence of the examiners on the result was not possible, because there were so many: each examiner tested only very few participants; in some cases, only one participant, so including the factor “examiner” as a factor in the ANOVA was not feasible. This is now mentioned in the ‘Limitations section’. Furthermore, an additional table with exact numbers of participants per examiner is now included in supplementary materials.

Here are the following edits which were introduced:

“Unfortunately, a direct test of the influence of the examiners on the result was not possible, because of the large number of examiners. Each examiner tested very few participants; some only one participant. Therefore, including examiners as a factor in the ANOVA was not feasible. Numbers of participants per each examiner are included in S8 Table. ” 

Results

Clear and full reporting of the results, including interaction effects and their meaning.

I miss information about the clustering of participants over examiner (how many per examiner?). Following the 'statistics' comments - it would be of added benefit to add a quantification of the effect of examiner/setting, if the author group agrees. If not, just a bit of info on the clustering would be enough.

Response: Thank you for this comment. Please see our response above re: adding the information on clustering.

Table 1. I see no added benefit of 2 decimal places in the scores - one is enough to maintain precision, but will increase overview and interpretation (e.g., less distraction). Generally, I'd suggest adding only one decimal if the raw measurement was without decimals, as these presented constructs are. For the catagorical, add % for quick interpretation.

Response: Thank you very much for your remark. However, if the reviewer agrees, we would prefer to stick with 2 decimals, which is a standard practice, especially considering future meta-analyses which require as precise data as possible. Percentages were added for categories in Table 1. 

Table 3. The addition of *p < .05 is not useful, since all are <.001 and ***.

Response: Table 3. Was corrected in terms of p-value reporting. Also, new correlations coefficients are reported in line with suggestion of reviewer #2.

r319 – even though it is interesting, the use of the word 'interestingly' suggest a contrast to the sentence before and is as such a bit confusing for me. Next / Further would be more pure for me.

Response: Word “Interestingly” was corrected to “Further”.

Discussion

Generally: The order of the presented discussion is not consistent with the order of the aims/intro/methods/results -> Here, the expectation is the primary aim, in the intro the reproduction is the first presented aim, in the methods and results, it's the effect itself (segement vs Ssp effect, then followed by a time effect).

Response: Thank you for your feedback. The order of the discussion was changed to reflect the order of the study aims. 

r 345; the order of the aims written here are not consistent with the introduction, methods and results.

Response: The first section of the discussion was rewritten, and sections of the discussion were moved to be consistent with aims in introduction 

r 366; 'significant in 9/10 cases' -> in the result (table III), all cases are highly significant?

Response: Thank you for this comment. Indeed, it was a mistake. It was corrected to significant in all cases.

r429; to consider (as commented in the introduction as well): I don't think its about spatial summation during pandemic - its about spatial summation outside of the lab what is the most important message here (which is discussed in this paragraph as well, as it should). I truly hope we will not have such a pandemic anymore, but I do hope with this innovative design we can do robust research in non-lab settings (which is also necessary sometimes for more translational studies). I'd suggest rephrasing the heading to "Spatial summation outside a laboratory environment" (or something like that).

Response: Thank you for your comment. We agree that the topic of this section should exceed beyond the pandemic. We rephrased the heading to: Spatial summation in a non-laboratory setting.

Paragraph r429 - I miss the discussion specifically about the home environment and differences between the setting/examiners (as commented on earlier). Future research could greatly benefit on this methodology, but it has to be discussed specifically. As it is now, only the 'why' and the method (used in all settings) is discussed (as it is adapted to less expensive equipment), but not the challenges that could be encountered. Please add a deeper discussion about the home environment AND the different settings used due to having multiple examiners/environments.

Response: The discussion section has been revised for relevance, application of the new methodology and possible difficulties arising from its use. A limitations section has also been added to emphasize the difficulties associated with environmental factors and differences between examiners. Please see introduced changes here:

Discussion: “Nonetheless, it is important to note a number of difficulties in applying such a methodology. The first one is maintaining the target temperature by adjusting the volume of CPT (water and ice) which must be preceded by testing. It is advisable to compare the resulting temperature with the available specialized CPT, as the authors have done (Skalski, et al., 2021). Another is the problem with standardizing the environment in which the assessment is conducted. The environment outside the lab does not provide as much control over the environmental stimuli as laboratory setting does. For this reason, research under such conditions can be more challenging and require very strict procedure of conduct.”

Limitations and future directions: “There are two main limitations to this study. First is the lack of standardization of the environmental factors that result from conducting this study in outside the laboratory settings. Examiners conducted the study in their households, resulting in significant differences in the characteristics of the selected rooms and environmental stimuli. The second is that the study was conducted by multiple examiners, differing in factors such as age and gender that can affect the subjects. Another is that the examiners varied significantly in the number of subjects assessed which comes as the result of restrictions during COVID-19 pandemic. Unfortunately, a direct test of the influence of the examiners on the result was not possible, because of the large number of examiners. Each examiner tested very few participants; some only one participant. Therefore, including examiners as a factor in the ANOVA was not feasible. Numbers of participants per each examiner are included in Appendix 8. Future attempts to adapt methodologies using non-laboratory equipment should emphasize standardizing the environment in which measurements take place to control environmental stimuli.”

Reviewer #2

According to the public registration, the study of Nastaj and colleagues aimed to answer the following questions :

-Can spatial summation of pain be observed in a home-based environment?

-Are pain expectations related to the perceived pain intensity during the spatial summation paradigm?

-Does the temperature of the water used for the spatial summation paradigm confound the spatial summation effect?

Even though it is not clearly stated as a hypothesis, the authors also aimed at assessing the replicability of the order effect observed by Marchand and Arsenault.

I think that the data presented in the manuscript can answer all these questions.

However, I have some general reservations on the interpretation that is made of some of the observed effects and more specific comments on parts of the manuscript. Also, I think that the manuscript would benefit from an extra round of proof-reading as some typos are still present.

Response: Thank you very much for your feedback and suggestions which improved our manuscript.

Expectations

I think there is some confusion on the treatment of expectations and their effects on pain in the manuscript.

The Authors show that the reports of expected and perceived pain are correlated. In my opinion, this shows that the participants have a good internal model of their nociceptive system and of the effect of stimulation surface area on the intensity of their pain sensation (spatial summation). This shows that participants expect larger stimuli to be more painful (in a way, an effect of spatial summation on expectations).

This, however, does not show an effect of expectations on spatial summation. To show that the spatial summation effect is modulated by expectation, the Authors should have manipulated the expectations of the participant and observed a change in perceived intensity congruent with the expectation manipulation, similarly to the studies that the authors cite in the introduction and discussion sections. Citations of these studies in conjunction with ambiguous phrasing of the effects observed by the authors in their own data makes the discussion slightly misleading.

Response: Thank you very much for your thoughtful comments. We do agree with the reviewer. To investigate the effect of expectations on SSp, we would have to include a manipulation of expectations. We investigated the relationship between expectations and SSp, but indeed some sentences could imply that we investigated the effects of expectancy manipulation. To address this, we rephrased some of the core sentences in the manuscript to make clearer that we investigated the association between expectations and pain induced by different body areas (segments). The purpose of studies we cited in the introduction is to underline the importance of studying expectations and their role in pain perception. However, as expectancy was never investigated regarding SSp (in any context), we first wanted to see if there were any links between expectations and SSp. Indeed, we also agree with the reviewer’s interpretation that our participants seemed to have a good internal model of their nociceptive system. Please see introduced changes here:

Abstract: “An additional aim was to explore the association between expectations and SSp”

Introduction: “ Furthermore, the current experiment aimed to investigate the potential association between pain-related expectations and SSp by examining participants’ predictions regarding pain using an SSp paradigm”

Discussion: “The last aim was to use this novel methodology to investigate the associations between expectations and SSp (…)”

Limitations and future directions: “Further research should focus on the manipulation of expectations to see if the subjects’ level of expectations affects SSp and is able to modulate it.”

Sensitization

For each immersion, perceived pain was reported three times: after 10, 30 and 50 s. Pain ratings are shown to increase between these time points. The Authors interpret this effect of time as sensitization. I don’t think that sensitization is the most likely explanation for this effect.

I am almost certain that (at least some of) this effect can be explained by continuously decreasing temperature at the depth of the afferents.

The temperature at the depth of the nociceptors is not the same as the temperature at the surface of the skin, it is a compromise between the temperature of the surface of the skin and the deep body temperature. Following a change of the surface temperature, the temperature at the depth of the afferents will (slowly, as heat needs to be conducted through tissues largely made of water) drift towards a new equilibrium. Given that cold nociceptors are thought to be located deeper in the skin than most nociceptors and given the large temperature delta caused by the immersion, the temperature at the depth of these nociceptors can be expected to significantly differ at 10, 30 and 50 s post immersion.

Additionally, temporal summation can be expected to play a role to the effect of time on pain intensity.

Response: Thank you for this comment and for pointing out other mechanisms to explain the observed results. Indeed, temporal summation seems to be a valid explanation for our observed results than sensitization. We rephrased parts of the manuscript related to sensitization and focused on the explanation of our results based on temporal summation. Please see introduced changes here:

Abstract: “This study suggests that SSp is shaped by a mixture of excitatory and inhibitory mechanisms and can be influenced by the temporal summation of the nociceptive system”

Results: “Furthermore, a significant main effect was found for the factor “time” (Wald �2(2) = 157.45, p < 0.001), indicating that temporal summation occurred during immersion”

Discussion: “Moreover, SSp is strongly influenced by temporal summation”.

Physiological mechanisms of SSp:“The curve representing the summation trajectory was steeper when the last measurements were considered (50s), indicating that SSp can be partially mediated by the temporal and/or spatial summation of nociceptors innervating deeper layers of soft tissue (Campero & Bostock, 2009) and possible veins (Klement & Arndt, 1992)”.

Conclusion: “Secondly, it seems that SSp can be shaped by a mixture of excitatory and inhibitory mechanisms and is influenced by the temporal summation of the nociceptive system”

“Lastly, spatial summation seems to be strictly linked with expectations, but future studies that directly modulate expectations are needed to investigate their influence on SSp, and physiological measurements should also be taken”

Descending pain inhibition

The order of presentation of the different stimuli (increasing or decreasing sequence) influenced the reported level of pain, at least for the smallest surface of stimulation. The Authors interpret this as evidence for a modulation of the spatial summation effect by descending inhibitory mechanisms.

Even though I don’t think that it is completely unlikely, I also don’t find the evidence compelling, and I think that the Authors reach this conclusion without sufficient argumentation. I would therefore suggest that they revise the discussion to reflect the thought process that led them to this conclusion.

Additionally, I think that the Authors should also consider the possibility that this order effect does not reflect a true difference in pain but is rather an artefact of measurement. Previous stimuli/perceptions can influence the way the participants use a rating scale. For this reason, evaluation of spatial summation effects should actually not be done with an ordered sequence such as the one used in this manuscript but rather with a random ordering of the different surfaces. Of course, as the authors aimed at replicating the order effect, using an ordered sequence made sense for their study.

Response: Thank you for your feedback. We agree that a random sequence of immersion of different surfaces would be better to evaluate and estimate “a clear” SSp. We aimed to replicate the methodology used in the Marchand & Arsenault study (2002) on other similar studies on SSp such as Julien & Marchand (2006). We based our argumentation on the results of the Marchand & Arsenault study (2002) in which the authors investigated three conditions: increasing session (immersion from fingertip to the axilla), decreasing session (immersion from the axilla to fingertip) and whole arm + increasing session (first immersion to the axilla and next from fingertip to the axilla). Their reasoning that inhibitory mechanisms were activated when the largest area was immersed at the beginning of the session was supported by their results, as SSp occurred in both, the decreasing session and the whole arm + increasing session, despite subsequent increasing immersions in the whole arm + increasing session. We agree that the discussion section was lacking sufficient argumentation on this issue, so we expanded “Reproducibility of spatial-related effects” to address this issue. 

Please see introduced changes here:

“Their findings demonstrated that SSp was only detected in the decreasing session and whole-arm-first + increasing session and did not take place in the increasing session condition. The authors proposed an explanation that in the increasing condition, both facilitatory and inhibitory mechanisms were gradually being recruited. The results from the third condition (whole-arm-first + increasing session) appear to support their suggestion. If the gradual recruitment of inhibitory mechanisms is responsible for the lack of correlation between the stimulated area and reported pain during the increasing session, a positive relationship between the stimulated area and pain should be reestablished in a whole-arm-first + increasing session whereas starting with the whole arm immersion would stimulate the activation of inhibitory mechanisms at the beginning of the session (…)” 

We also agree that previous stimuli, perceptions, and simple predictions can influence the way participants use a rating scale. However, in the CPT immersion paradigm, this issue is very difficult to solve as participants always see or feel the size of the immersed hand which can influence their rating even if order of immersion is random. 

SPECIFIC POINTS

-“Spatial summation of pain (SSp) can be defined as an increase of pain intensity when body area (1–3) or distance (4,5) between stimulated areas are increased (3).” (41-42)

The definition of spatial summation put forward by the authors is not the conventional definition. It misses the fact that spatial summation affects not only the perceived intensity of suprathreshold stimuli but also the threshold. Additionally, the distance between two stimuli modulates spatial summation but is not spatial summation per se and the relationship between stimulus distance and perceived intensity is not as simple as described here (at some point the effect saturates or even reverse).

Response: Thank you for your comment. We rephrased this section of the introduction to address this issue:

“Spatial summation of pain (SSp) may be characterized as an increase of pain intensity when body area (Adamczyk, et al., 2021; Julien & Marchand, 2006; Marchand & Arsenault, 2002) or distance ( Adamczyk, et al., 2021, Quevedo & Coghill, 2009) between stimulated areas are enlarged (Marchand & Arsenault, 2002). There is also evidence that pain thresholds decrease as the area of noxious stimulation increases (Defrin et al.,2003; Lautenbacher et al., 2001; Defrin & Urca, 1996). However, studies show that when the area of noxious stimulation or the distance between stimulated areas exceeds 30 cm (in case of cold pain) (Defrin et al., 2011) or 10 cm (in case of heat pain) (Defrin et al.,2006,) SSp does not occur.”

-“[…] is that there is no linear increase in pain intensity during a linear increase in the stimulated body area (1) or in the distance between the stimuli (1,4), further suggesting that inhibitory processes are involved in SSp.” (45-47)

I am not sure I understand why the authors think the non-linearity of the stimulus surface - perceived intensity suggests that inhibitory processes are involved. Most stimulus intensity – perception functions are non-linear. Additionally, the non-linearity of this relationship is not obvious based on the data presented in this manuscript.

Response: Thank you for your remark re: stimulus-response data. Indeed, most of the psychophysical data are nonlinear in the context of pain. For instance, traditional stimulus-response functions of pain ratings and stimulus intensity show exponential increase. These data are very robust in the context of noxious heat, mechanical stimuli, electrical stimuli. However, sometimes pain does not increase exponentially. Instead, it can follow a logarithmic pattern if the intensity of stimulation is constant but the surface of stimulation is manipulated (Adamczyk et al., 2021). We refined the meaning of this sentence to reflect the nature of the increase in pain (decay):

“An interesting observation is that there is no exponential or linear increase in pain intensity during a linear increase in the stimulated body area (Adamczyk et al., 2021a) or in the distance between the stimuli (Adamczyk et al., 2021a, Adamczyk et al., 2021b). Instead, an increase in pain seems to be characterized by a relatively quick saturation when the area of the body exposed to noxious stimulation is expanded in a linear fashion, further suggesting that inhibitory processes are involved in SSp.”

What we think happened in this study is, that the stimulus surface was not increasing linearly like in one of our previous reports wherein the stimulated surface increased linearly causing a disproportional increase in pain. In the current study, the pain increase was not saturated when a greater number of segments were immersed, which we think is because the area of the body increased more rapidly (exponentially). This finding could be echoing more advanced facilitation than inhibition.

-“The low temperature activates nociceptive fibers (13,17,18) through low temperature sensitive ion channels TRPM8” (49-50)

The identity of the noxious cold receptors remains a matter of debate. Aside from TRPM8, TRPA1 is also considered as a good candidate and there is evidence suggesting that a third channel could be involved.

Response: Thank you for your feedback. We corrected this sentence. Information about TRPA1 channels is now included in 

introduction. 

“The low temperature activates nociceptive fibers (Wolf & Hardy, 1941; Campero, Serra, & Ochoa, 1996; Middleton et al., 2021) through low temperature sensitive ion channels TRPM8 (McKemy, Neuhausser, & Julius, 2002) and TRPA1(Kozyreva, Kozaruk, & Meyta, 2019)” 

-“CPT is a suitable method to study SSp; the stimulated bodily area can be adjusted in a controlled and standardized manner by 

immersion depth (2,15).” (51-52)

I agree that CPT can be used to study spatial summation. However, I think that this sentence minimizes the fact that this approach does not actually allow to control the exact stimulation surface area, as every participant’s anatomy is different.

Response: Thank you for your comment. We agree that participant’s anatomy could play a role in controlling the exact stimulation surface area. To address this, we rephrased this sentence: 

“Although this method does not allow adequate standardization when there are anthropometric differences, it is easy and commonly used and has been successfully employed in pain studies.”.

-“The size of areas exposed to cold stimulation is presented in Figure 2.” (138-139)

It is not clear to me how this size was measured. I don’t think it is currently explained in the manuscript. I think it is important to describe it.

Response: Thank you for your feedback. Indeed, it was not mentioned in the manuscript. To address this issue a sentence was added to the methods section: 

“Segment lines were marked on the skin with a pen. Length and width of each segment was measured using flexible measuring tape before the main phase of experiment.”

“In the first stage, the effect of stimulation area on pain intensity was investigated using a General Estimated Equations (GEE) model with three within subject factors: “condition” (ascending, descending), “segment” (1, 2, 3, 4, 5 segments) “time” (10s, 30s, 50s).” (187-189)” - Where the different levels of factors “condition”, segment”, and “time” treated as factors or continuous variables?

Response: We would like to thank the reviewer for this question. Different levels of factors were treated as “factors”. Please see the modified version of this sentence:

“In the first stage, the effect of stimulation area on pain intensity was investigated using a General Estimated Equations (GEE) model with three within-subject conditions treated as “factors” : “condition” (ascending, descending), “segment” (1, 2, 3, 4, 5 segments) “time” (10s, 30s, 50s).”

In the supplementary materials, the authors indicate that they decided to use a GEE instead of a GLM (as per registration) because the pain ratings were not normally distributed. Non normality of the outcome variable is not a violation of the assumptions of the GLM. Non normality of the residuals (the difference between the values predicted by the models and the true data) is. If the authors did, indeed, not try fitting a GLM because the pain ratings were not normally distributed, I think they should fit such a model and then assess the normality of the residuals. If those are indeed normally distributed, the authors should report the results of the GLM rather than the GEE. That way, they would deviate less from their registration.

Response: Thank you very much for this important analytical remark. We have run the traditional repeated-measure GLM first, and also tested the normality of standardized and non-standardized residuals. Not only the measured outcomes did not follow normal distribution but also residuals deviated in this regard (all Shapiro-Wilk tests {30 in total} and 27/30 Kolmogorov tests indicated violation). We thus decided to keep the results and analysis as they are (slightly more conservative approach). Of note, traditional GLM showed similar pattern of results.

Similarly, normality of the data is not a condition to use Pearson’s correlation and I think the Authors should stick to the metric they initially chose. As the existence of a linear relationship is a condition to use this metric, I would recommend including scatter plots of the data along the correlation coefficients, so that the reader can verify that the relation is approximate linear.

Response: Thank you for your comment. We have now re-analyzed the correlations following the reviewer’s suggestion and calculated Pearson’s product-moment correlation coefficients consistently. See changes reflected here:

“Pearson coefficients (r = 0.53 - 0.81) were statistically significant for all correlations between expected pain and actual pain (Table 3)” 

We also provided scatter plots to visualize associations between predicted (internally modelled) pain and actual pain of participants (Figure 5, raw data)

In parallel to what I mentioned in the previous paragraph, it is good practice to include visualization of the raw data and not just the marginal means/CI in the figures.

Response: Thank you for your comment. We kept descriptive statistics based on raw data whereas on figures we showed estimated means as the analysis of statistical significance applies to estimated rather, not raw means. Having both within the paper ensures that the reporting is completed. We also added the following:

“Unless otherwise indicated figures and tables include descriptive statistics based on raw data.”

“Mean temperature of the cold water was 5.12°C (��0.47) during the ascending condition (t[66] = -0.5, p = 0.63) as well as during the descending condition (t[66] =1.2, p = 0.22) no significant differences (before condition vs. after condition) were shown for the pain thresholds, indicating that pain sensitivity was stable over the course of experiment (Appendix 4).” - I don’t think that these analyses are described in the method section. I think they should.

Response: The reviewer is correct. These analyses were not initially described. We have now added them to the analysis section:

“The comparison between water temperature and PPTs before and after the experiment was conducted using paired Student t tests.

- “Descriptive statistics for expectation are presented in Figure 3 and Table 2.” (284) - Are the plotted values marginal means or sample means? The text sometimes indicates one think and sometimes the opposite.

Response: We apologize for this confusion we now made this detail clear and reported if estimated means are presented or not.

“Unless otherwise indicated figures and tables include descriptive statistics based on raw data.”

-“No difference between expectancy measured before the first immersion (unbiased) versus expectancy measured on a trial-by-trial basis were found (Wald �2(1) = 1.52, p = 0.22), thus it was decided to perform main expectancy analysis on the pooled dataset.” (284-287) - I don’t think that this analysis has been described in the method section. I think it should.

Response: Indeed, this analysis was not originally included in methods section. We have now added this to appropriate section:

“Also, two sets of expectations ratings (measured before vs. after) were compared within the same statistical model to explore potential differences ”

Additionally, I don’t think that the absence of a statistical difference is a good reason to pool the data and I would like the before trial and during trial expectancy data to be analyzed separately, as pooling may obscure an effect of the previous stimulus on expected for the next stimulus.

Response: We do not think that pooling had a negative influence on the data, considering that the distribution of this type of measure were comparable. However, we analyzed them separately and present them now within the manuscript and supplementary materials in line with the suggestion:

“No significant difference was observed between expectations measured before the first immersion (unbiased) and expectations measured during the assessment (Wald X2(1) = 1.52, p = 0.22). Thus, for clarity reasons, expectations measured during the assessment are presented below whereas measures taken prior to assessment are in S7 Table.

-“Interestingly, the repeated measures ANOVA of the size of immersed segments showed a significant effect for “size”, indicating that the area of stimulation increased from trial to trial (F(1,67) = 135.17, p < 0.001, �2 p = 0.70).” (320-321)

I don’t think that this analysis has been described in the method section. I think it should.

Response: Indeed, this analysis was not originally included in methods section. We have now added this to appropriate section:

“Next, we analyzed differences in the actual sizes of areas of stimulation (approximated via anthropometric measurements) by using repeated measures ANOVA and size of the segment (1 to 5) as dependent variable.”

Reviewer #3

The present study investigates new aspects of SSp by exploring the involvement of expectations. At the same time, this study replicates previous results, slightly changing the experimental design; given the replication crisis, replicating previous data must be valued. An additional novelty of this study is the introduction of a new feasible way to induce SSp using a home-based set-up. The experiment is methodologically sound and leads to novel findings, including the association between expectations and perceived pain in SSp. The manuscript is generally well-written and clear. The discussion could benefit from revision and reorganisation, empathising which results are the most novel and which ones are not. Some additional details should be added in the analysis and in the results section. Overall, this is an interesting and thoroughly designed work.

Response: We would like to sincerely thank the reviewer for their time and effort given to review our paper. Please find below our responses to your comments in a point-by-point fashion.

Abstract

Line 30-33. These two novel findings should be highlighted in the discussion. Here it is clear that these two findings are a novelty, while in the discussion (i) is not presented as a novelty. In the abstract, I suggest mentioning novelty ii- expectation) first, and then the other novelty (I - spatial summation and time), because the investigation of expectation is one of the primary aims of the study.

Response: Thank you very much for rising this valid paint. Now findings regarding expectations are mentioned as first. Please see introduced changes here: 

“Furthermore, two novel findings were observed: i) there was a significant correlation between expectations and perceived pain, indicating a link between pain expectations and SSp, ii) spatial summation increased with the increase in duration exposure to the noxious stimulus (χw(2) = 157.5, p < 0.001).”

We also underline ii) finding as novelty in discussion section. Please see introduced changes here:

“Our results suggest that the duration of the stimulus strongly affects the summation pattern. The curve representing the summation trajectory was steeper when the last measurements were considered (50s), indicating that SSp can be partially mediated by the temporal and/or spatial summation of nociceptors innervating deeper layers of soft tissue (Campero & Bostock, 2009) and possible veins (Klement & Arndt, 1992)”. 

Line 32- 33. ‘indicating that pain expectations can contribute to SSp’. Either replace ‘can’ with ‘could’ or rephrase the sentence with something like: indicating a link between expectations and SSp.

Response: Sentence was corrected to “indicating a link between expectations and SSp”.

Design and method

Please specify whether the examiners physically went to the participants’ houses or whether the participants physically went to the examiners’ houses. Please clarify this point also in relation to the covid situation (was it during a time in which it was possible to go to each other houses but not in the lab?). Or maybe I misunderstood, and it was done online? Please clarify this point.

Response: Thank you for your comment. Although we have written in the paper that participants were assessed within examiners households. We did not provide a specific information in the paper about how this assessment looked like. To address this we added a sentences in “General information”:

“The study was conducted in a home environment, i.e., each examiner performed an experiment within their households (see details below). The home setting differed between examiners. Mainly living rooms, bedrooms and kitchens were chosen to build up the setup. Settings were chosen such that the subject could stay without distractions such as music or the sounds of other household members.”

113 Remove comma after ‘fact’

Response: Comma was removed.

Figure 1. Since there is a drawing for the ascending session, I suggest adding the drawing for the descending one (although this is not necessary).

Response: Thank you for your feedback. The descending session was identical to the ascending, they differed in the size of area immersed first. We decided to show the descending session in order to show readers the method in a clear way. The main reason we did not include descending drawing was to not add figures that would take up space and not contribute much.

Just by reading the Experimental Design section, it is not clear whether the immersion of each hand section is sequential with breaks between each segment test, or whether each segment test occurs one after the other, without breaks. The experiment design becomes clearer when reading the ‘Trial Design’ section in which this information is specified. However, it would be best if this structure was clearer from earlier on in the manuscript. I suggest adding a few sentences that clarify this also in the Experimental Design section, while further details are then given in the ‘Trial Design’. Please also add a reference that backs up the choice of using a 5 minutes break between each trial as a sufficient time to reach the baseline temperature level.

Response: Thank you for comment. To address that aspect, we have now added the following sentence to “experimental design” section: 

“In both conditions there was a 5 minutes break between immersion of the next segment to ensure that the skin temperature reached a baseline level (Adamczyk et al., 2022). The order of testing (ascending vs. descending condition) was random. The interval between each condition was one hour”. 

158 ‘Before and after each experimental condition, the cold pressor threshold was tested’. Was this done before and after each segment? Or just before and after the ascending and the descending. Please clarify. If the cold pressor threshold was not measured before each trial, this could be mentioned as a potential limit.

Response: The cold pain threshold was tested before and after two conditions. We decided to not test cold pain threshold before and after each segment. We assumed that it could interfere with recovering of skin temperature to baseline level. To address this issue and clarify it in manuscript we rephrase this sentence to:

“Before and after each experimental condition (ascending, descending), cold pain thresholds (PTCOLD) were tested on the examined limb: an ice cube was placed on the palmar surface of the participants’ hand and the time (in seconds) until the first pain sensation occurred, was recorded ( Tilley & Bisset, 2017)”.

182 I would replace ‘conduct’ with ‘run’

Response: The word “conduct” was changed to “run”.

Results

226 was the temperature identical for the descending and the ascending? 5.12°C

Response: The temperatures were as follows: 5.13°C during ascending condition and 5.12°C during descending condition. Sentence is now corrected to:

“The mean temperature of the cold water was 5.13°C during the ascending condition and 5.12°C during the descending condition”.

Clarification is required, both in the statistical analysis (line 244) and in the results section, regarding the Bonferroni-corrected p-value. Please specify if the p-values reported in the manuscript are already corrected. For each interaction, please also specify the number of planned comparisons and how this was used to correct for Bonferroni. My understanding is that for each interaction there are 5 comparisons (comparison 1: 1/5 ascendent vs 1/5 ascendent - comparison 2: 2/5 ascendent vs 2/5 ascendent - comparison 3: 3/5 ascendent vs 3/5 ascendent - comparison 4: 4/5 ascendent vs 4/5 ascendent - comparison 5: 5/5 ascendent vs 5/5 ascendent), if this is true the corrected p-value threshold should be 0.01. If the corrected p-value is 0.01, some of the significance of some results should be rediscussed (es. p=0.017 for the ‘condition’ x ‘area’ x ‘time would be almost significant. Given that Bonferroni is very strict, this should not be a problem but should still be acknowledged). However, I might be missing something here, and the p-values might have already been corrected (i.e., multiplying the p-value by 5). Please, clarify this aspect for all the interaction effects (es. ‘Condition’ x ‘segment’; ‘segment’ x ‘time’; ‘condition’ x ‘area’ x ‘time’.

Response: We now applied the Sidak correction instead of Bonferroni because Sidak is slightly less conservative. All p-values reported are already adjusted. Also, we corrected for type I-error rate rather than adjusting p-values associated with main and interaction effects which is not a standard practice. Overall results remained unchanged.

Line 130-131. The authors state that anger measurements have been taken. However, these data are not included in the analysis. Please, explain why this interesting variable has not been considered. If no association between fear of pain and pain has been found, I would still recommend mentioning this as an insignificant result and adding a brief comment on why this might be the case (e.g., would be interesting to check whether fear of pain is a predictor of perceived intensity at the first trial. Individuals with higher fear of pain might experience greater pain in the first pain test. The first trial could be interesting because, given the novelty of the experience, this might be more influenced by fear).

Response: We appreciate the reviewer’s comments and suggestions. We assumed that this is just a typo because we measured fear instead of anger (although we agree that measuring emotional states more comprehensively in the future studies on spatial summation is a totally underexplored area of research). We also did some exploratory correlations in this matter. We did not find any significant correlations of fear of pain (general pain) and pain in the first trial nor pain thresholds (p > 0.05).

Discussion

Line 353 – 354 the authors state ‘that spatial summation i) can be inferred from subjects’ pain expectations and ii) is strongly influenced by sensitization, iii) is likely shaped by descending pain inhibition.’ While these are important points, the link between each point and the findings of this study are not always clear throughout the discussion. I suggest making it very clear in the discussion how the results of this study support each of these claims.

Response: Thank you for this comment. We discuss this finding in separate sections of discussion. We rephrased this sentence to be more cohesive with discussion structure: 

“Furthermore, our results contribute to the mechanistic understanding of SSp by showing that spatial summation might be predicted from subjects’ pain expectations. Moreover, SSp is strongly influenced by temporal summation. Additionally, differential effects of the order of immersions indicate that -to some degree- SSp is shaped by descending pain inhibition”.

Line 387. Before 'Despite' need full stop.

Response: Full stop was added before “Despite”.

394 immersion of segment 1 was slightly ‘higher’? Maybe I am missing something, but my understanding is that pain perception of segment 1 in the descending condition is ‘lower’.

Response: Thank you for this valid point. Indeed, it was an editing mistake. Now this sentence was corrected:

“It was observed that when the stimulation started from the largest area, pain provoked by immersion of segment 1 was slightly lower than the analogue stimulation in the ascending sequence”.

In the section ‘Physiological mechanisms of SSp’ the distinction between facilitatory and inhibitory processes and how these are linked to your results could be clearer. My understanding is that from line 393 to line 399 the authors are talking about inhibitory mechanisms, while from line 399 to line 407 the authors discuss facilitatory responses. Please, make this distinction clearer.

Response: Thank you very much for this comment. We separately discuss inhibition and facilitation in the context of spatial summation here:

“The mechanisms of SSp are not fully understood, however, it seems that both facilitatory and inhibitory processes interact during summation. Inhibitory processes could be inferred from an interaction between the sequence and the stimulated body area (segments). It was observed that when the stimulation started from the largest area, pain provoked by immersion of segment 1 was slightly lower than the analogue stimulation in the ascending sequence. That discrepancy can be explained by the fact that in the descending sequence, inhibitory mechanisms are activated to their maximum and persist during subsequent immersions, thereby resulting in lower pain during the immersion of segment 1, which is in line with a previous study (Marchand & Arsenault, 2002). Moreover, a study on rats which showed that an increase in stimulation area from 1.9 to 18cm2 gradually decreased the frequency of convergent neuron discharges in intact, yet not spinally transected animals (Bouhassira et al.,1995). Furthermore, the increase in pain in either sequence was disproportional. A 5 times larger area does not multiply the reported pain intensity by the same value, which is in line with previous SSp studies (Adamczyk et al., 2021;Marchand & Arsenault, 2002;Julien et al., 2005). 

As for facilitatory processes, it appears that they could be modulated by other factors. Temporal summation has not yet been considered in SSp induced via CPT. Our results suggest that the duration of the stimulus strongly affects the summation pattern”.

A limitation and future directions section could be added.

Response: Thank you for this valid point. Indeed, manuscript missed that section. Section of limitations and future direction was added.

Conclusions

Line 456. Please keep the hierarchy of the aims fixed across the manuscript. At lines 345 -348 the aims are well stated, I recommend keeping this hierarchy throughput the text (i.e., also in the abstract and in the introduction, in which the order is, at times, mixed). Since one of the main aims was to replicate the previous findings with a different design, I suggest mentioning this in the conclusion.

Response: We restructured this section to be cohesive with introduction. 

Line 457 ‘were emerged’: remove ‘were’

Response: The word “were” has been removed. 

Line 457 – 459. In the present study expectations were shown to predict pain perception, however, these were not modulated and therefore no conclusion can be drawn on the potential effects of expectations on pain. I suggest writing something on the lines of: Firstly, spatial summation seems to be strictly linked with expectations, but future studies that directly modulate expectations are needed to investigate their influence on SSp, and physiological measurements should also be measured.

Response: Thank you for your comment. We rephrased it to: 

“Lastly, spatial summation seems to be strictly linked with expectations, but future studies that directly modulate expectations are needed to investigate their influence on SSp, and physiological measurements should also be taken”.

 Line 459 ‘confirmed’ becomes ‘confirm’

Response: Please see respons above as sentence was changed. 

 

References

Adamczyk, W. M., Katra, M., Szikszay, T. M., Peugh, J., King, C. D., Luedtke, K., & Coghill, R. C. (2023). Spatial Tuning in Nociceptive Processing Is Driven by Attention. The Journal of Pain, 24(6), 1116-1125. 

Adamczyk WM, Manthey L, Domeier C, Szikszay TM, Luedtke K. Nonlinear increase of pain in distance-based and area-based spatial summation. Pain. 2021a Jun;162(6):1771–80.

Adamczyk WM, Szikszay TM, Kung T, Carvalho GF, Luedtke K. Not as “blurred” as expected? Acuity and spatial summation in the pain system. Pain. 2021b Mar;162(3):794–802.

Bouhassira D, Gall O, Chitour D, Bars DL. Dorsal horn convergent neurones: negative feedback triggered by spatial summation of nociceptive afferents. Pain. 1995 Aug;62(2):195–200.

Büchel C, Geuter S, Sprenger C, Eippert F. Placebo Analgesia: A Predictive Coding Perspective. Neuron. 2014 Mar;81(6):1223–39.

Campero M, Bostock H. Unmyelinated afferents in human skin and their responsiveness to low temperature. Neurosci Lett. 2010 Feb;470(3):188–92

Campero M, Serra J, Ochoa JL. C-polymodal nociceptors activated by noxious low temperature in human skin. J Physiol. 1996 Dec 1;497(2):565–72. 

Coghill RC, Yarnitsky D. Healthy and normal? The need for clear reporting and flexible criteria for defining control participants in quantitative sensory testing studies. Pain. 2015 Nov;156(11):2117–8

Carlino E, Benedetti F. Different contexts, different pains, different experiences. Neuroscience. 2016 Dec 3;338:19-26. doi: 10.1016/j.neuroscience.2016.01.053. Epub 2016 Jan 28. .

Colloca L, Lopiano L, Lanotte M, Benedetti F. Overt versus covert treatment for pain, anxiety, and Parkinson's disease. Lancet Neurol. 2004 Nov;3(11):679-84.

Defrin R, Givon R, Raz N, Urca G. Spatial summation and spatial discrimination of pain sensation. Pain. 2006 Dec;126(1):123–31.

Defrin R, Ronat A, Ravid A, Peretz C. Spatial summation of pressure pain: effect of body region. Pain. 2003 Dec;106(3):471–80. 

Defrin R, Sheraizin A, Malichi L, Shachen O. Spatial summation and spatial discrimination of cold pain: Effect of spatial configuration and skin type. Pain. 2011 Dec;152(12):2739–45. 

Defrin R, Urca G. Spatial summation of heat pain: a reassessment. Pain. 1996 Jul;66(1):23–9. 

Julien N, Goffaux P, Arsenault P, Marchand S. Widespread pain in fibromyalgia is related to a deficit of endogenous pain inhibition. Pain. 2005 Mar;114(1):295–302

Julien N, Marchand S. Endogenous pain inhibitory systems activated by spatial summation are opioid-mediated. Neurosci Lett. 2006 Jul;401(3):256–60. 

Klement W, Arndt JO. The role of nociceptors of cutaneous veins in the mediation of cold pain in man. J Physiol. 1992 Apr 1;449(1):73–83

Kozyreva TV, Kozaruk VP, Meyta ES. Skin TRPA1 ion channel participates in thermoregulatory response to cold. Comparison with the effect of TRPM8. J Therm Biol. 2019 Aug;84:208–13.

Lautenbacher, Jesper Nielsen, Thim S. Spatial summation of heat pain in males and females. Somatosens Mot Res. 2001 Jan;18(2):101–5. 

Marchand S, Arsenault P. Spatial summation for pain perception: interaction of inhibitory and excitatory mechanisms. Pain. 2002 Feb;95(3):201–6.

McKemy DD, Neuhausser WM, Julius D. Identification of a cold receptor reveals a general role for TRP channels in thermosensation. Nature. 2002 Mar;416(6876):52–8.

Middleton SJ, Barry AM, Comini M, Li Y, Ray PR, Shiers S, et al. Studying human nociceptors: from fundamentals to clinic. Brain. 2021 Jun 22;144(5):1312–35

Miki Y, Chubachi N, Imamura F, Yaegashi N, Ito K. Impact of COVID-19 restrictions on the research environment and motivation of researchers in Japan. Prog Disaster Sci. 2020 Dec;8:100128. 

Moseley GL, Arntz A. The context of a noxious stimulus affects the pain it evokes. Pain. 2007 Dec 15;133(1-3):64-71. doi: 10.1016/j.pain.2007.03.002. Epub 2007 Apr 20.

Quevedo AS, Coghill RC. Filling-In, Spatial Summation, and Radiation of Pain: Evidence for a Neural Population Code in the Nociceptive System. J Neurophysiol. 2009 Dec;102(6):3544–53.

Skalski J, Nastaj J, Swoboda S, Budzisz A, Zbroja E, Małecki A, et al. Nielaboratoryjna adaptacja badania przestrzennego sumowania bólu w dobie pandemii COVID-19. BÓL. 2021;21:1–7.

Sohrabi C, Mathew G, Franchi T, Kerwan A, Griffin M, Soleil C Del Mundo J, et al. Impact of the coronavirus (COVID-19) pandemic on scientific research and implications for clinical academic training – A review. Int J Surg. 2021 Feb;86:57–63.Coghill RC, Yarnitsky D. Healthy and normal? The need for clear reporting and flexible criteria for defining control participants in quantitative sensory testing studies. Pain. 2015 Nov;156(11):2117–8

Song Y, Yao M, Kemprecos H, Byrne A, Xiao Z, Zhang Q, et al. Predictive coding models for pain perception. J Comput Neurosci. 2021 May;49(2):107–27.

Tilley P, Bisset L. The Reliability and Validity of Using Ice to Measure Cold Pain Threshold. BioMed Res Int. 2017;2017:1–6.

Uman LS, Stewart SH, Watt MC, Johnston A. Differences in High and Low Anxiety Sensitive Women’s Responses to a Laboratory‐Based Cold Pressor Task. Cogn Behav Ther. 2006 Dec;35(4):189–97. 

Vigil JM, Rowell LN, Alcock J, Maestes R. Laboratory Personnel Gender and Cold Pressor Apparatus Affect Subjective Pain Reports. Pain Res Manag. 2014;19(1):e13–8.

Wiech K. Deconstructing the sensation of pain: The influence of cognitive processes on pain perception. Science. 2016 Nov 4;354(6312):584–7..

Wolf S, Hardy JD. STUDIES ON PAIN. OBSERVATIONS ON PAIN DUE TO LOCAL COOLING AND ON FACTORS INVOLVED IN THE “COLD PRESSOR” EFFECT. J Clin Invest. 1941 Sep 1;20(5):521–33.

---

## [Decision Letter · Decision Letter 1]

10 Nov 2023

PONE-D-23-02600R1Spatial summation of pain is associated with pain expectations: Results from home-based paradigmPLOS ONE

Dear Dr. Adamczyk,

Thank you for submitting your manuscript to PLOS ONE. After careful consideration, we feel that it has merit but does not fully meet PLOS ONE’s publication criteria as it currently stands. Therefore, we invite you to submit a revised version of the manuscript that addresses the points raised during the review process.

The manuscript has been evaluated by three reviewers, and their comments are available below.

The reviewers are positive about the work and the revision but have raised a number of concerns that need  additional attention. Please find their detailed comments below. 

We look forward to receiving your revised manuscript.

Kind regards,

Vanessa Carels

Staff Editor

PLOS ONE

Reviewers' comments:

Reviewer's Responses to Questions

**Comments to the Author**

1. If the authors have adequately addressed your comments raised in a previous round of review and you feel that this manuscript is now acceptable for publication, you may indicate that here to bypass the “Comments to the Author” section, enter your conflict of interest statement in the “Confidential to Editor” section, and submit your "Accept" recommendation.

Reviewer #1: All comments have been addressed

Reviewer #2: (No Response)

Reviewer #3: All comments have been addressed

2. Is the manuscript technically sound, and do the data support the conclusions?

Reviewer #1: Yes

Reviewer #2: Partly

Reviewer #3: Yes

3. Has the statistical analysis been performed appropriately and rigorously? 

Reviewer #1: Yes

Reviewer #2: Yes

Reviewer #3: Yes

4. Have the authors made all data underlying the findings in their manuscript fully available?

Reviewer #1: Yes

Reviewer #2: Yes

Reviewer #3: Yes

5. Is the manuscript presented in an intelligible fashion and written in standard English?

Reviewer #1: Yes

Reviewer #2: Yes

Reviewer #3: Yes

6. Review Comments to the Author

Reviewer #1: Dear authors,

The manuscript as it is presented now seems to be much more clear and and the readability is improved a lot. Moreover, the message (and evidence) of conducting these experiments outside the lab (but with strict procedures and reporting of possible influential factors) is conveyed much better and will hopefully help the academic world to progress/translate into the clinic, give oppertunities to researches without such a lab and hopefully enstrenghten our understanding of pain further to help our patients in pain.

As a minor last remark, COVID-19 as keyword doesn't seem appropiate in my opinion. Possibly the editorial office can erase it, since the study doesn't study COVID, and we preferrably don't want to give 'noise' to systematic review searchers on the subject.

Finally, I want to thank the authors for their great work!

Roland Reezigt

Reviewer #2: I thank the Authors for their responses to my comments. I think the manuscript as greatly improved since the previous stage and is almost ready for publication. I however still have a few remarks, exposed hereunder.

Main points:

-The Authors state that “Spatial summation of pain (SSp) may be characterized as an increase of pain intensity when [… the] distance [4,5] between stimulated areas are enlarged [3].” The distance between stimulated areas has been shown to modulate spatial summation, something referred to as “lateral inhibition” by certain authors, but this is not spatial summation proper, a concept with a long history which has always been about larger stimulation surface area. I would therefore invite the Authors to rephrase this sentence. Changing the definition of spatial summation may put the pain literature at odds with the haptics and thermosensation literature.

-“An interesting observation is that there is no exponential or linear increase in pain intensity during a linear increase in the stimulated body area [1] or in the distance between the stimuli [1,4]. Instead, an increase in pain seems to be characterized by a relatively quick saturation when the area of the body exposed to noxious stimulation is expanded in a linear fashion, further suggesting that inhibitory processes are involved in SSp.”

I am a bit confused by these statements as the classically described stimulus-response function in psychophysics is concave-down increasing, which led Fechner to propose a logarithmic psychophysical law in the 19th century. Stevens later showed that Fechner was wrong and that a power law should be used instead but this only marginally affects the shape of the function, not the orientation of the concavity. The observations of the Authors are therefore perfectly in line with the psychophysical literature but the introduction seems to suggest otherwise.

Additionally, this type of behaviour could be completely accounted for by partially overlapping receptive fields at the different levels of the neural hierarchy and I therefore think that the Authors should be more cautious when designating inhibitory mechanisms as a cause for this phenomenon.

-I appreciate the fact that the Authors enthusiastically adopted my suggestion of temporal summation as a mechanism explaining the difference in spatial summation between the time points but I think a more cautious style should be adopted when mentioning it as it is only a putative mechanism that could account for part of the effect. On the other hand, I think more emphasis should be put on the fact that, due to the slowness of heat conduction through the skin, from the point of view of the primary afferent (i.e. at the depth of the receptors), the temperature must have been quite different at the different time-points. This could be expected to drive a large part of the temporal effect observed by the Authors but is only briefly mentioned in the discussion.

-“In our study, this interaction was replicated, although it was prominent when only the immersion of the first segment (fingertips) was compared.” I think that a more nuanced approach is needed here. In the study of Marchand and Arsenault, almost no spatial summation was observed in the increasing condition whereas a very clear pattern of spatial summation was observed in the decreasing condition. In the case of the present study, pain ratings are largely colinear with an effect similar to that reported by Marchand and Arsenault but of much smaller magnitude only apparent at the 50 s time point. I would say that the replication was partially successful, which does not reduce the value of the Author’s study which nicely shows that this effect seems there but is a bit more subtle than what could have been expected from the Marchand and Arsenault paper. This discrepancy may come from the different type of stimulus (noxious cold vs noxious heat) something that could be discussed (heat pain tends to arise faster than cold pain, which could maybe explain why the effect seemed present only at the later time points).

Off note, when looking at the “slope” of the rating-surface relationship for time point 50s, there seems to be an overall reduction of spatial summation for the ascending condition (flatter slope), something that is maybe not entirely capture by the current data analysis (as different surfaces are modelled as factors) but goes in the direction of the Authors hypothesis.

Minor points:

Abstract:

“The influence of psychological factors, such as the volunteer’s expectations of pain intensity, on the actual perception of pain were also measured on a VAS.” For the reasons explained during the previous round, I think this sentence is misleading as the Author’s showed a relationship/correlation between predicted and perceived intensity but not an influence of the former on the later. I therefore recommend rephrasing the sentence accordingly.

Introduction:

-“The low temperature activates nociceptive fibers [18,22,23] through low temperature sensitive ion channels TRPM8 [24] and TRPA1 [25] leading to increasing pain of mild to moderate intensity [26].” As I mentioned in my previous comments, TRPA1 may be a transducer of noxious cold but this is uncertain. I think this uncertainty should be reflected in the phrasing, e.g. “probably through low temperature sensitive ion channels TRPM8 [24] and/or TRPA1”.

-“Although this method does not allow adequate standardization when there are anthropometric differences, it is easy and commonly used and has been successfully employed in pain studies.” I think the Authors are too hard on themselves, standardization is not perfect but I think it is adequate enough.

- “Both publication bias, based on phenomenon of higher probability of publishing statistically significant findings than nonsignificant findings [33], and lack of replication studies are affecting the validity of scientific research [34]. This problem was particularly addressed by researchers in the field of psychology, leading to “crisis of confidence” in previous findings [35], but it is also found in other fields like medicine [36], economy, [37] or genetic research [38].” It is up to the Authors to decide but I don’t personally think that it is necessary to introduce the reproducibility crisis to justify an attempt at replication.

Methods:

-“Length and width of each segment was measured using flexible measuring tape before the main phase of experiment.” Thanks for adding this information. I am however still not sure of how these measures were transformed into surface areas. Did the Authors model the segments as rectangular parallelepipeds?

-“The comparison between water temperature and PPTs before and after the experiment was conducted using paired Student t tests” I don’t think the PPT abbreviation is defined.

Discussion:

-“Inhibitory processes could be inferred from an interaction between the sequence and the stimulated body area (segments).” By sequence, do the Authors mean ascending/descending?

-This sentence seems to be missing a verb. “Moreover, a study on rats which showed that an increase in stimulation area from 1.9 to 18cm2 gradually decreased the frequency of convergent neuron discharges in intact, yet not spinally transected animals [52].”

Reviewer #3: I thank the authors for having integrated the manuscript with the given feedback. In my opinion, the manuscript has increased its quality and clarity. Below are some minor revisions. A general comment is that the authors could give a last round of revision, checking the grammar and the language clarity.

Line 70. Typo. ‘To address this IS in SSp’. The authors state one of their aims in this sentence. However, this is not so clear. I suggest stating this aim more clearly (e.g., To address this in SSp, we aim to replicate this effect, adopting the methodology from the study of Marchand & Arsenault, in the present study’.

Line 78. ‘Furthermore, the current experiment…’ I would not put this in a new paragraph. I would continue in the same paragraph so that the two aims are clearly stated in one paragraph.

Spelling mistakes in S4 ‘after 60 seconds hands is of the cold water 5 minutes breakv’

Line 443. ‘Another is that’ Paraphs missing a word ‘Another limitation is that’

452. ‘Further research should focus on the manipulation of expectations to see if the subjects' level of expectations affects SSp and is able to modulate it.’ Good observation. I suggest making this comment more explicit saying that the present study is limited to the associative level (as indicated by correlations), while modulating expectations could give insights on the causal relationship between expectancy and SSp.

459-460. I suggest writing a separate sentence for the comment on physiological measurements.

7. PLOS authors have the option to publish the peer review history of their article (what does this mean?). If published, this will include your full peer review and any attached files.

Reviewer #1: **Yes: **Roland R Reezigt

Reviewer #2: No

Reviewer #3: **Yes: **Eleonora Maria Camerone

---

## [Author Response · Author response to Decision Letter 1]

7 Dec 2023

Reviewer #1

The manuscript as it is presented now seems to be much more clear and the readability is improved a lot. Moreover, the message (and evidence) of conducting these experiments outside the lab (but with strict procedures and reporting of possible influential factors) is conveyed much better and will hopefully help the academic world to progress/translate into the clinic, give opportunities to researches without such a lab and hopefully strengthen our understanding of pain further to help our patients in pain.

Response: Thank you very much for your positive feedback and suggestions which improved our manuscript.

As a minor last remark, COVID-19 as keyword doesn't seem appropriate in my opinion. Possibly the editorial office can erase it, since the study doesn't study COVID, and we preferably don't want to give 'noise' to systematic review searchers on the subject.

Response: Thank you for remark. We removed “COVID-19” as a keyword.

Finally, I want to thank the authors for their great work!

Response: Thank you so much for your time to review our manuscript!

Reviewer #2

I thank the Authors for their responses to my comments. I think the manuscript as greatly improved since the previous stage and is almost ready for publication. I however still have a few remarks, exposed hereunder.

Response: We are very thankful for your positive feedback and further set of comments on our study

Main points:

-The Authors state that “Spatial summation of pain (SSp) may be characterized as an increase of pain intensity when [… the] distance [4,5] between stimulated areas are enlarged [3].” The distance between stimulated areas has been shown to modulate spatial summation, something referred to as “lateral inhibition” by certain authors, but this is not spatial summation proper, a concept with a long history which has always been about larger stimulation surface area. I would therefore invite the Authors to rephrase this sentence. Changing the definition of spatial summation may put the pain literature at odds with the haptics and thermosensation literature.

Response: Thank you for your feedback. We rephrased part of introduction referring to definition of spatial summation and spatial 

conditions under which it occurs: 

“Spatial summation of pain (SSp) is characterized by an increase of perceived pain intensity when the area of nociception is enlarged [1–3]. Furthermore, pain thresholds decrease as the area of noxious stimulation increases [4–6] which can also be seen as a spatial summation manifestation. There are also studies indicating that SSp can occur when the area of nociception is not contiguous [4,5]. However, studies show that when the separation between stimulated areas exceeds 30 cm (in case of cold pain) [7] or 10 cm (in the case of heat pain) [8], SSp no longer occurs”

-“An interesting observation is that there is no exponential or linear increase in pain intensity during a linear increase in the stimulated body area [1] or in the distance between the stimuli [1,4]. Instead, an increase in pain seems to be characterized by a relatively quick saturation when the area of the body exposed to noxious stimulation is expanded in a linear fashion, further suggesting that inhibitory processes are involved in SSp.” I am a bit confused by these statements as the classically described stimulus-response function in psychophysics is concave-down increasing, which led Fechner to propose a logarithmic psychophysical law in the 19th century. Stevens later showed that Fechner was wrong and that a power law should be used instead but this only marginally affects the shape of the function, not the orientation of the concavity. The observations of the Authors are therefore perfectly in line with the psychophysical literature but the introduction seems to suggest otherwise. Additionally, this type of behavior could be completely accounted for by partially overlapping receptive fields at the different levels of the neural hierarchy and I therefore think that the Authors should be more cautious when designating inhibitory mechanisms as a cause for this phenomenon.

Response: Thank you for this valid point. We would like to clarify the meaning of this cited sentence. We do not refer to sensory experience magnitude which correspond with stimulus intensity (magnitude), but to experience magnitude corresponding to the “size” of the stimulated area. What is interesting in these early psychophysical studies is that percept increased exponentially (positive exponent). However, pain seems to rather increase logarithmically when the stimulus intensity is constant, but the area increases linearly – this is what we were referring to. Along the same lines, percept seems to be more linear when the area or “size” of the stimulation increases more exponentially, like in the current study. We changed the language in the following part of 

the introduction:

“An interesting observation is that there is no exponential increase in pain intensity during a linear increase in the “size” of the stimulated area [1], a typical stimulus-response pattern when the stimulus are (size) is constant but the noxious intensity increases linearly [13,14].”

-I appreciate the fact that the Authors enthusiastically adopted my suggestion of temporal summation as a mechanism explaining the difference in spatial summation between the time points but I think a more cautious style should be adopted when mentioning it as it is only a putative mechanism that could account for part of the effect. On the other hand, I think more emphasis should be put on the fact that, due to the slowness of heat conduction through the skin, from the point of view of the primary afferent (i.e. at the depth of the receptors), the temperature must have been quite different at the different time-points. 

Response: Thank you for your feedback. We rephrased some parts of the manuscript related to temporal summation: 

Abstract: “This study suggests that SSp is associated with pain expectations and can be formed by a mixture of excitatory and inhibitory mechanisms potentially driven by temporal characteristics of neural excitation”

Discussion: “Moreover, SSp can be potentially driven by temporal characteristics of neural excitation”.

We also rephrased part of the discussion to emphasize role of temperature:

Physiological mechanisms of SSp:

“The curve representing the summation trajectory was steeper when the last measurements were considered (50s). During the immersion in cold water, the hand temperature had to change with time, hence, observed SSp could be driven by the gradual activation of deeply located primary afferents. This may indicate that SSp can be partially mediated by temporal and/or spatial summation of nociceptors innervating deeper layers of soft tissues [54] and possibly deep veins [55]”.

-“In our study, this interaction was replicated, although it was prominent when only the immersion of the first segment (fingertips) was compared.” I think that a more nuanced approach is needed here. In the study of Marchand and Arsenault, almost no spatial summation was observed in the increasing condition whereas a very clear pattern of spatial summation was observed in the decreasing condition. In the case of the present study, pain ratings are largely colinear with an effect similar to that reported by Marchand and Arsenault but of much smaller magnitude only apparent at the 50 s time point. I would say that the replication was partially successful, which does not reduce the value of the Author’s study which nicely shows that this effect seems there but is a bit more subtle than what could have been expected from the Marchand and Arsenault paper. This discrepancy may come from the different type of stimulus (noxious cold vs noxious heat) something that could be discussed (heat pain tends to arise faster than cold pain, which could maybe explain why the effect seemed present only at the later time points).

Off note, when looking at the “slope” of the rating-surface relationship for time point 50s, there seems to be an overall reduction of spatial summation for the ascending condition (flatter slope), something that is maybe not entirely capture by the current data analysis (as different surfaces are modelled as factors) but goes in the direction of the Authors hypothesis.

Response: Thank you for this valid point. We extended the discussion section on differences between our and Marchand & Arsenault (2002) study. We also calculated regression coefficients in order to directly compare the findings to result from 2002 (especially, slopes and intercepts). We added following changes:

Data extraction and statistical analysis:

“Linear regression was calculated to describe the relationship between pain and number of stimulated segments – similarly to the described procedures in the Marchand & Arsenault study [3]”

Results:

“ (…) and regressions coefficients: Slope of pain increase was steeper in descending compared to ascending condition, however, the most noticeable difference in steepness was found for pain measured at 50s (B = 9.45 vs. 6.20, see Table S9)”

We also emphasized some key differences between the current study and experiment from 2002:

Discussion:

“These differences include the localization of noxious stimulation as well as the overall size of the stimulated surface area. Only the hand was used for the current study and was divided into 5 segments, while Marchand & Arsenault [3] used the entire arm and divided it into 8 segments. Secondly, the type of noxious stimulation was different. Here, noxious cold stimulation was used, compared to noxious heat stimulation in the previous report. The type of noxious stimuli could explain the results as the most noticeable difference of pain increase between the two conditions was observed after 50s of immersion. This could be a result of temporal summation of noxious cold pain, which arises slower compared to noxious heat pain [52]. Thirdly, the study was conducted with home-based equipment outside of the laboratory.”

Minor points:

Abstract:

“The influence of psychological factors, such as the volunteer’s expectations of pain intensity, on the actual perception of pain were also measured on a VAS.” For the reasons explained during the previous round, I think this sentence is misleading as the Author’s showed a relationship/correlation between predicted and perceived intensity but not an influence of the former on the later. I therefore recommend rephrasing the sentence accordingly.

Response: Thank you for pointing this out. Sentence in abstract was rephrased to: 

“Psychological factors, such as the participants’ expectations of pain intensity were also measured on a VAS.” 

Introduction:

-“The low temperature activates nociceptive fibers [18,22,23] through low temperature sensitive ion channels TRPM8 [24] and TRPA1 [25] leading to increasing pain of mild to moderate intensity [26].” As I mentioned in my previous comments, TRPA1 may be a transducer of noxious cold but this is uncertain. I think this uncertainty should be reflected in the phrasing, e.g. “probably through low temperature sensitive ion channels TRPM8 [24] and/or TRPA1”.

Response: Thank you for your feedback. Word “probably” was added to the sentence.

-“Although this method does not allow adequate standardization when there are anthropometric differences, it is easy and commonly used and has been successfully employed in pain studies.” I think the Authors are too hard on themselves, standardization is not perfect, but I think it is adequate enough.

Response: Thank you for this comment. Sentence was rephrased to:

 “Although standardization in this method is difficult due to e.g., anthropometric differences, it is easy and commonly used and has been successfully employed in pain experiments”

- “Both publication bias, based on phenomenon of higher probability of publishing statistically significant findings than nonsignificant findings [33], and lack of replication studies are affecting the validity of scientific research [34]. This problem was particularly addressed by researchers in the field of psychology, leading to “crisis of confidence” in previous findings [35], but it is also found in other fields like medicine [36], economy, [37] or genetic research [38].” It is up to the Authors to decide but I don’t personally think that it is necessary to introduce the reproducibility crisis to justify an attempt at replication.

Response: We fully agree that there is no need to mention replication crisis to justify the replication attempt. However, still there is a negative bias towards conducting and publishing replication studies in many scientific fields. So, we decided to include this part in our paper, however, we shorten it significantly.

Methods:

-“Length and width of each segment was measured using flexible measuring tape before the main phase of experiment.” Thanks for adding this information. I am however still not sure of how these measures were transformed into surface areas. Did the Authors model the segments as rectangular parallelepipeds?

Response: Yes, however based on 2 dimensions as this was the easiest and the fastest approximation. During the immersion, the participants were instructed to keep their hands straight. In such position shape of the hand is similar to a parallelogram tapering slightly toward the tips of the fingers. We collected measurements of two dimensions of each segment: length (a) and width (b). Then these values were multiplied to obtain an approximated surface area for each segment (ax × bx). Following sentence was added to the manuscript: 

“To calculate the “area” of immersed segments, surface areas (ax × bx) of each immersed segment were summed. For example: the surface area of 5/5 segments was calculated as: first segment (a1 × b1) + second segment (a2 × b2) + third segment (a3 × b3) + fourth (a4 × b4) fifth segment (a5 × b5).”

-“The comparison between water temperature and PPTs before and after the experiment was conducted using paired Student t tests” I don’t think the PPT abbreviation is defined.

Response: Thank you very much for pointing this out. PPT was incorrect word, we meant PTCOLD. We clearly missed it. Sentence is now rephrased to: 

“The comparison between water temperature and PTCOLD before and after the experiment was conducted using paired Student t tests.”

Discussion:

-“Inhibitory processes could be inferred from an interaction between the sequence and the stimulated body area (segments).” By sequence, do the Authors mean ascending/descending?

Response: Thank you very much for pointing this out. Correct, what we meant was a sequence of particular immersions. Sentence is now rephrased to: “Inhibitory processes could be inferred from an interaction between the ascending and descending immersion and the stimulated body area (segments).”

-This sentence seems to be missing a verb. “Moreover, a study on rats which showed that an increase in stimulation area from 1.9 to 18cm2 gradually decreased the frequency of convergent neuron discharges in intact, yet not spinally transected animals [52].”

Response: Thank you for this comment. Sentence was corrected to: “Moreover, one study conducted on rats showed that an increase in stimulation area from 1.cm2 9 to 18cm2 gradually decreased the frequency of convergent neuron discharges in intact, yet not spinally transected animals”

Reviewer #3

I thank the authors for having integrated the manuscript with the given feedback. In my opinion, the manuscript has increased its quality and clarity. Below are some minor revisions. A general comment is that the authors could give a last round of revision, checking the grammar and the language clarity.

Response: Thank you very much for your feedback and for taking time to review our manuscript.

Line 70. Typo. ‘To address this IS in SSp’. The authors state one of their aims in this sentence. However, this is not so clear. I suggest stating this aim more clearly (e.g., To address this in SSp, we aim to replicate this effect, adopting the methodology from the study of Marchand & Arsenault, in the present study’.

Response: Thank you for pointing this out. Sentence was rephrased to: “To address this for the topic of SSp, one of the aims for the current study was to replicate the SSp effect, adopting the methodology from the study published by Marchand & Arsenault [3].”

Line 78. ‘Furthermore, the current experiment…’ I would not put this in a new paragraph. I would continue in the same paragraph so that the two aims are clearly stated in one paragraph.

Response: Thank you for your feedback. Sentence is now in the same paragraph. 

Spelling mistakes in S4 ‘after 60 seconds hands is of the cold water 5 minutes breakv’

Response: Thank you for this comment. S4 was corrected. 

Line 443. ‘Another is that’ Paraphs missing a word ‘Another limitation is that’

Response: Thank you for your feedback. Word “limitation” was added to the sentence. 

452. ‘Further research should focus on the manipulation of expectations to see if the subjects' level of expectations affects SSp and is able to modulate it.’ Good observation. I suggest making this comment more explicit saying that the present study is limited to the associative level (as indicated by correlations), while modulating expectations could give insights on the causal relationship between expectancy and SSp.

Response: Thank you very much for your suggestions. This comment was rephrased to: 

“The present study is limited to making inferences at the association level, rather than establishing causation. Further research should focus on experimentally manipulated expectations to test if subjects' expectations affect SSp. This could give an insight on the causal relationship between expectancy and SSp”.

459-460. I suggest writing a separate sentence for the comment on physiological measurements.

Response: Thank you for this comment. Separate sentence on physiological measurements was added: 

“Additionally, the inclusion of physiological measures could prove beneficial in future studies.”

References

1. Adamczyk, W. M., Katra, M., Szikszay, T. M., Peugh, J., King, C. D., Luedtke, K., & Coghill, R. C. (2023). Spatial Tuning in Nociceptive Processing Is Driven by Attention. The Journal of Pain, 24(6), 1116-1125.

2. Adamczyk WM, Manthey L, Domeier C, Szikszay TM, Luedtke K. Nonlinear increase of pain in distance-based and area-based spatial summation. Pain. 2021a Jun;162(6):1771–80.

3. Adamczyk WM, Szikszay TM, Kung T, Carvalho GF, Luedtke K. Not as “blurred” as expected? Acuity and spatial summation in the pain system. Pain. 2021b Mar;162(3):794–802.

4. Campero M, Bostock H. Unmyelinated afferents in human skin and their responsiveness to low temperature. Neurosci Lett. 2010 Feb;470(3):188–92

5. Compte, A., & Wang, X. J. (2006). Tuning curve shift by attention modulation in cortical neurons: a computational study of its mechanisms. Cerebral Cortex, 16(6), 761-778.

6. Defrin R, Givon R, Raz N, Urca G. Spatial summation and spatial discrimination of pain sensation. Pain. 2006 Dec;126(1):123–31.

7. Defrin R, Sheraizin A, Malichi L, Shachen O. Spatial summation and spatial discrimination of cold pain: Effect of spatial configuration and skin type. Pain. 2011 Dec;152(12):2739–45.

8. Hoeppli, M. E., Nahman-Averbuch, H., Hinkle, W. A., Leon, E., Peugh, J., Lopez-Sola, M., ... & Coghill, R. C. (2022). Dissociation between individual differences in self-reported pain intensity and underlying fMRI brain activation. Nature communications, 13(1), 3569.

9. Julien N, Marchand S. Endogenous pain inhibitory systems activated by spatial summation are opioid-mediated. Neurosci Lett. 2006 Jul;401(3):256–60. 

10. Klement W, Arndt JO. The role of nociceptors of cutaneous veins in the mediation of cold pain in man. J Physiol. 1992 Apr 1;449(1):73–83

11. Marchand S, Arsenault P. Spatial summation for pain perception: interaction of inhibitory and excitatory mechanisms. Pain. 2002 Feb;95(3):201–6.

12. Reid, E., Harvie, D., Miegel, R., Spence, C., & Moseley, G. L. (2015). Spatial summation of pain in humans investigated using transcutaneous electrical stimulation. The Journal of Pain, 16(1), 11-18

13. Quevedo AS, Coghill RC. Filling-In, Spatial Summation, and Radiation of Pain: Evidence for a Neural Population Code in the Nociceptive System. J Neurophysiol. 2009 Dec;102(6):3544–53. 

14. Quevedo AS, Mørch CD, Andersen OK, Coghill RC. Lateral inhibition during nociceptive processing. Pain. 2017 Jun;158(6):1046–52.

15. Stevens, S. S. (1970). Neural Events and the Psychophysical Law: Power functions like those that govern subjective magnitude show themselves in neurelectric effects. Science, 170(3962), 1043-1050.

16. Streff, A., Kuehl, L. K., Michaux, G., & Anton, F. (2010). Differential physiological effects during tonic painful hand immersion tests using hot and ice water. European journal of pain, 14(3), 266-272.

---

## [Decision Letter · Decision Letter 2]

27 Dec 2023

Spatial summation of pain is associated with pain expectations: Results from a home-based paradigm

PONE-D-23-02600R2

Dear Dr. Adamczyk,

We’re pleased to inform you that your manuscript has been judged scientifically suitable for publication and will be formally accepted for publication once it meets all outstanding technical requirements.

Kind regards,

Tomoyoshi Komiyama, Ph.D

Academic Editor

PLOS ONE

Additional Editor Comments (optional):

Dear Authors,

Thank you for submitting your revised manuscript.

Your research proposed new tools for SSp to be assessed outside the laboratory, such as in clinical settings. You also showed that SSp appears to be formed by a mixture of excitatory and inhibitory mechanisms and influenced by the temporal summation of nociceptive systems. Additionally, you suggest that although spatial summation appears to be related to expectations, future studies that directly adjust for expectations are needed to investigate its effects on SSp.

I think it was much easier to understand than the original manuscript.

I am satisfied with the responses and the edits, I am happy to accept this manuscript.

The authors have replied to our remaining questions satisfactorily from three reviewers.

Therefore, I have no further comments to make, all of my previous concerns were adequately addressed.

This manuscript will be satiating the reader's interest.

Tomoyoshi Komiyama

Reviewers' comments:

Reviewer's Responses to Questions

**Comments to the Author**

1. If the authors have adequately addressed your comments raised in a previous round of review and you feel that this manuscript is now acceptable for publication, you may indicate that here to bypass the “Comments to the Author” section, enter your conflict of interest statement in the “Confidential to Editor” section, and submit your "Accept" recommendation.

Reviewer #2: All comments have been addressed

Reviewer #3: All comments have been addressed

2. Is the manuscript technically sound, and do the data support the conclusions?

Reviewer #2: Yes

Reviewer #3: Yes

3. Has the statistical analysis been performed appropriately and rigorously? 

Reviewer #2: Yes

Reviewer #3: Yes

4. Have the authors made all data underlying the findings in their manuscript fully available?

Reviewer #2: Yes

Reviewer #3: Yes

5. Is the manuscript presented in an intelligible fashion and written in standard English?

Reviewer #2: Yes

Reviewer #3: Yes

6. Review Comments to the Author

Reviewer #2: (No Response)

Reviewer #3: (No Response)

7. PLOS authors have the option to publish the peer review history of their article (what does this mean?). If published, this will include your full peer review and any attached files.

Reviewer #2: No

Reviewer #3: **Yes: **Eleonora Maria Camerone

---

## [Editor Report · Acceptance letter]

23 Jan 2024

PONE-D-23-02600R2 

PLOS ONE

Dear Dr. Adamczyk, 

I'm pleased to inform you that your manuscript has been deemed suitable for publication in PLOS ONE. Congratulations! Your manuscript is now being handed over to our production team.

Kind regards, 

on behalf of

Dr. Tomoyoshi Komiyama 

Academic Editor

PLOS ONE